# A combined quantitative mass spectrometry and electron microscopy analysis of ribosomal 30S subunit assembly in *E. coli*

**Dipali G Sashital[1†‡], Candacia A Greeman[1†], Dmitry Lyumkis[1,2¶], Clinton S Potter[1,2], Bridget Carragher[1,2], James R Williamson[1,3,4]***

[1]Department of Integrative Structural and Computational Biology, Scripps Research Institute, La Jolla, United States; [2]National Resource for Automated Molecular Microscopy, Scripps Research Institute, La Jolla, United States; [3]Department of Chemistry, Scripps Research Institute, La Jolla, United States; [4]Skaggs Institute for Chemical Biology, Scripps Research Institute, La Jolla, United States

**Abstract** Ribosome assembly is a complex process involving the folding and processing of ribosomal RNAs (rRNAs), concomitant binding of ribosomal proteins (r-proteins), and participation of numerous accessory cofactors. Here, we use a quantitative mass spectrometry/electron microscopy hybrid approach to determine the r-protein composition and conformation of 30S ribosome assembly intermediates in Escherichia coli. The relative timing of assembly of the 3′ domain and the formation of the central pseudoknot (PK) structure depends on the presence of the assembly factor RimP. The central PK is unstable in the absence of RimP, resulting in the accumulation of intermediates in which the 3′-domain is unanchored and the 5′-domain is depleted for r-proteins S5 and S12 that contact the central PK. Our results reveal the importance of the cofactor RimP in central PK formation, and introduce a broadly applicable method for characterizing macromolecular assembly in cells.

***For correspondence:** jrwill@ scripps.edu

[†]These authors contributed equally to this work

**Present address:** [‡]Roy J Carver Department of Biochemistry, Biophysics and Molecular Biology, Iowa State University, Ames, United States; [¶]Laboratory of Genetics, Salk Institute for Biological Studies, La Jolla, United States

**Competing interests:** The authors declare that no competing interests exist.

**Reviewing editor**: Nahum Sonenberg, McGill University, Canada

## Introduction

The ribosome catalyzes protein biosynthesis and is essential for cell growth. In *Escherichia coli* (*E. coli*), the 70S ribosome is a large (2.4 MDa) ribonucleoprotein consisting of a small (30S) and large (50S) subunit. Because ribosome biogenesis is complex and taxing on the metabolic resources of the cell, the process is tightly regulated. The efficiency of ribosome assembly is so directly tied to cell growth that even slight defects in assembly confer a significant selective disadvantage, and strong defects can threaten cell survival. In addition, ribosomes must be assembled accurately to ensure the fidelity of protein synthesis. A large number of accessory factors have been implicated in the regulation and efficiency of ribosomal production, although the precise roles for many of these factors remain unknown (Reviewed in *Wilson and Nierhaus, 2007*; *Shajani et al., 2011*).

Remarkably, the 30S subunit can be reconstituted in vitro from 16S ribosomal RNA (rRNA) and 20 ribosomal proteins (r-proteins) in a high temperature, high $Mg^{2+}$ environment (*Traub and Nomura, 1969*). Early work by Nomura and colleagues established the order and dependencies of r-protein binding in the assembling 30S subunit under equilibrium conditions (*Figure 1A*) (*Mizushima and Nomura, 1970*; *Held et al., 1973*, *1974*). In the early stages of assembly, primary r-proteins bind directly to the 5′-, central and 3′-domains of 16S rRNA. These initial r-protein binding events lead to changes in the rRNA structure, and facilitate subsequent binding of secondary and tertiary r-proteins

**eLife digest** The proteins in cells are made by complex organelles called ribosomes. These organelles are made of two subunits: the small ribosomal subunit, which reads the messenger RNA that contains the genetic code for the protein, and the large ribosomal subunit, which links amino acids together to form a protein. But how are the ribosomes themselves—which contain several ribosomal RNA molecules and dozens of ribosomal proteins—put together?

Various aspects of the assembly of ribosomes have been studied in the test tube, but the complexity of the assembly process means there is little data from experiments performed on living cells. Now Sashital et al. have used a combination of two techniques—mass spectrometry and electron microscopy—to study the assembly of ribosomes in living *Escherichia coli* cells. Mass spectrometry measures the relative amounts of the different ribosomal proteins in each sample, while electron microscopy provides information on the shape of the ribosome, including the shape of some of the intermediate structures formed during the assembly process.

Sashital et al. analyzed the composition and structure of the small ribosomal subunits in wild type *E. coli*, and also in mutant *E. coli* cells in which the genes for various proteins thought to be involved in the assembly process had been deleted. These experiments revealed that a protein called RimP had a key role in stabilizing an important central structure called a pseudoknot. The approach developed by Sashital et al. should be able to reveal other details about the assembly of ribosomes, and also about other macromolecular complexes that are found inside the cells.

(*Held et al., 1973*; *Stern et al., 1989*). More recent studies using time-resolved hydroxyl radical RNA structure probing (*Adilakshmi et al., 2008*), fluorescence correlation spectroscopy (*Ridgeway et al., 2012*), single-molecule fluorescence resonance energy transfer (*Kim et al., 2014*), pulse-chase monitored by quantitative mass spectrometry (*Talkington et al., 2005*; *Bunner et al., 2010a*), and time-resolved negative stain electron microscopy (*Mulder et al., 2010*) added valuable insight into the dynamics and kinetics of RNA folding, r-protein binding, and immature subunit conformations throughout the assembly process. These studies have revealed that even in the presence of r-protein binding dependencies, assembly can proceed through multiple parallel pathways. In addition, a large body of evidence indicates that misfolded rRNA structure leads to stable kinetic traps during in vitro 30S reconstitution, inhibiting the binding of several secondary and tertiary r-proteins and limiting the efficiency of the reconstitution (Reviewed in *Sykes and Williamson, 2009*).

In the cell, 30S assembly is fast and efficient, proceeding with the help of numerous assembly factors, including enzymes that directly modify the 16S rRNA and r-proteins, as well as a number of RNA-binding chaperones and GTPases that assist in RNA folding (Reviewed in *Wilson and Nierhaus, 2007*; *Shajani et al., 2011*). Previous studies suggest that some assembly factors, such as RimM and RbfA, may promote efficient assembly by binding co-transcriptionally to the nascent rRNA to facilitate folding and to prevent the formation of kinetic traps (*Williamson, 2003*; *Clatterbuck Soper et al., 2013*). In addition, factors may guide rearrangement of rRNA structure at specific points during the assembly process, as is likely the case during the re-structuring and cleavage of the 16S 5′-leader sequence late in assembly (*Dammel and Noller, 1995*). Cryo-EM reconstructions indicate that several assembly factors, including RbfA (*Datta et al., 2007*; *Jomaa et al., 2011b*), the GTPases Era (*Sharma et al., 2005*) and RsgA (*Guo et al., 2011*), and the 16S rRNA methyltransferase KsgA (*Boehringer et al., 2012*), bind immature subunits and block them from prematurely entering the translation cycle. Many assembly factors appear to be functionally related, forming a complex network of interconnected activities (*Bylund et al., 1998*; *Lu and Inouye, 1998*; *Inoue et al., 2003*; *Campbell and Brown, 2008*; *Goto et al., 2011*; *Connolly and Culver, 2013*). Other factors, such as RimP, have been implicated in 30S assembly but have no known connection to the overall assembly factor network (*Nord et al., 2009*; *Bunner et al., 2010b*).

One of the major obstacles hindering studies of in vivo ribosomal biogenesis stems from the complexity of the assembly process and heterogeneity of the incompletely assembled intermediates. Multiple parallel pathways are operative for assembly, giving rise to a variety of intermediates containing distinct sets of r-proteins (*Talkington et al., 2005*; *Mulder et al., 2010*). Quantitative mass spectrometry (qMS) provides a high-throughput method for precisely measuring the relative levels of

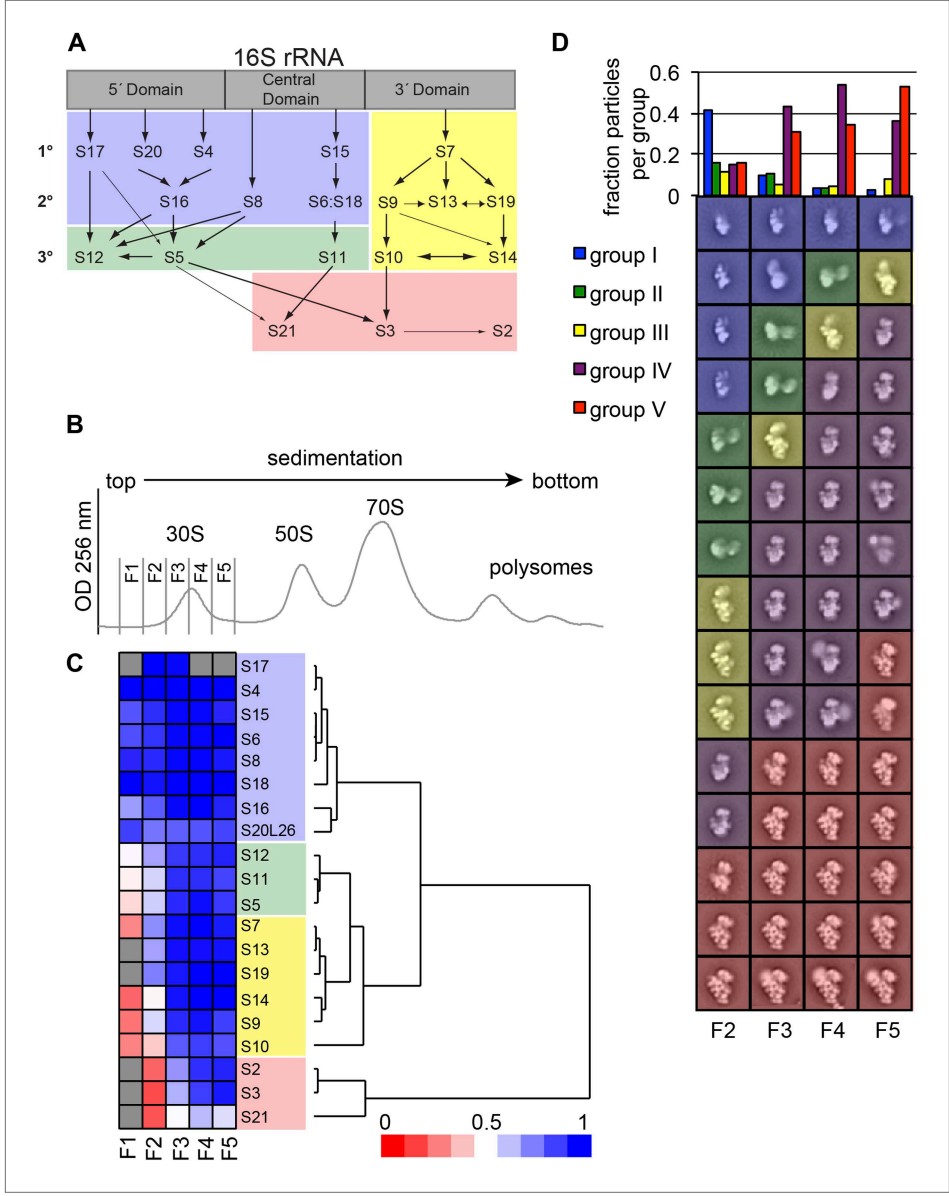

**Figure 1**. High-throughput qMS/EM analysis of assembly intermediates from WT *E. coli*. (**A**) Nomura assembly map; adapted. Three major regions of 16S rRNA are labeled at top. Arrows represent binding dependencies, with primary, secondary and tertiary proteins labeled on left. (**B**) Sucrose gradient chromatogram (absorbance at 254 nm) for WT *E. coli* lysate. 30S peak fractions analyzed by qMS and EM are labeled. (**C**) R-proteins clustered by relative abundance in 30S particles across sucrose gradient fractions 1–5 with a blue to red gradient representing high to low relative abundance. Relative abundance of each r-protein was normalized to that of S4. Gray boxes indicate r-proteins for which no peptides were detected. Clusters of r-proteins from more abundant to less abundant across the gradient are highlighted on the right as blue, green, yellow, and red. (**D**) Negative stain EM class averages for sucrose gradient fractions 2–5 (labeled at bottom). Classes were obtained by reference-free maximum likelihood alignment and classification and are sorted by Group. Histogram at top shows the fractional contribution of particles from each dataset to each Group.

The following figure supplements are available for figure 1:

**Figure supplement 1**. Raw micrographs and initial class averages for WT F2 negative stain EM data set.

**Figure supplement 2**. Hierarchical clustering of class averages.

proteins in a complex mixture (*Charollais et al., 2003*). Recently, qMS has been used to determine the composition of r-proteins within ribosomal assembly intermediates isolated from cells, revealing the binding dependencies of late-associating r-proteins on earlier binding r-proteins and assembly factors (*Chen and Williamson, 2013*; *Clatterbuck Soper et al., 2013*; *Guo et al., 2013*; *Leong et al., 2013*). Single-particle EM is also an ideal technique for analyzing heterogeneous samples due to the development of powerful alignment and classification schemes that enable the identification of sub-populations of particle conformations within a single sample (*Frank, 2006*). Automated data collection and downstream image processing has greatly improved the throughput of single-particle EM analysis, facilitating the rapid analysis of even highly heterogeneous samples in a short period of time (*Suloway et al., 2005*; *Lander et al., 2009*). In addition, automated random conical tilt (RCT) data collection enables three-dimensional reconstructions of each subpopulation identified in a sample (*Radermacher et al., 1987*; *Yoshioka et al., 2007*; *Voss et al., 2010*). These advances in automation were previously applied to study the in vitro assembly of the 30S subunit, resulting in a quantitative visualization of assembly intermediate conformations present at various stages of the in vitro reconstitution process (*Mulder et al., 2010*).

In order to visualize the distributions of 30S assembly intermediates present in the cell we have developed a hybrid approach combining qMS and single-particle EM to provide unprecedented insight into the composition and structure of ribosome assembly intermediates from cellular lysates. Our approach exploits the exquisite facility of both techniques to analyze the heterogeneous samples generated by fractionating crude *E. coli* lysates using sucrose gradient ultracentrifugation. This one-step sample preparation eliminates the use of affinity tags or purification steps that may inadvertently lead to the exclusion of early intermediates. For each gradient fraction, ribosomal protein levels were measured using qMS, revealing r-proteins present or depleted within assembly intermediates. In parallel, gradient fractions were analyzed using single-particle negative stain EM to elucidate the structures of assembly intermediates and mature ribosomes present in each sample. These techniques were used to analyze *E. coli* 30S subunit assembly in wild type (WT) cells and in several assembly factor deletion strains. These studies revealed a novel intermediate, where the central pseudoknot (PK) that connects the 5′-body domain of the 16S rRNA with the 3′-head domain, is unformed, although the 3′-head domain is partially assembled. Deletion of the assembly factor RimP causes a striking defect in central PK stability and results in the depletion of central PK-adjacent proteins S5 and S12, and late-binding proteins S2, S3 and S21. Together, our data suggest that central PK formation can occur either before or after head domain formation. Furthermore, our data implicate RimP in efficient central PK formation and in the subsequent incorporation of tertiary r-proteins S2, S3, S5, S12 and S21. In addition to providing novel insights into ribosome assembly, our approach represents a generalizable toolkit for studying the assembly of supramolecular structures in heterogeneous cellular samples.

## Results

### Observation of in vivo 30S assembly intermediates by qMS and negative stain EM

We previously demonstrated that ribosome assembly intermediates could be separated from mature subunits by sucrose gradient centrifugation (*Chen and Williamson, 2013*). Quantitative MS (qMS) analysis of sucrose gradient fractions revealed that individual r-protein levels in early 30S and early 50S fractions are variable and consistent with the expected binding dependencies based on the Nomura (*Held et al., 1974*) (*Figure 1A*) and Nierhaus (*Herold and Nierhaus, 1987*) maps, respectively. For example, in early 30S fractions, r-protein levels cluster into four groups with 5′-domain primary and secondary binders such as S4 and S8 in the most abundant group and late binders such as S2, S3 and S21 in the most depleted group. The presence of four distinct groups of r-protein levels suggests that several intermediate species with varied r-protein compositions are present in early 30S fractions. In order to investigate the subpopulations of intermediates that accumulate in vivo, we combined our qMS analysis with single particle EM, applied to fractions collected from the 30S peak.

*E. coli* BW25113 (Keio collection background strain/WT) (*Baba et al., 2006*) grown in M9 minimal media was harvested during exponential growth to ensure active production of ribosomes and the steady-state presence of assembly intermediates in the culture. The cell lysate was resolved by sucrose gradient centrifugation and five fractions encompassing the entire 30S peak were collected for qMS and EM analysis, allowing for a direct comparison of r-protein compositions with the observed particle conformations (*Figure 1B*). For qMS analysis, an equimolar amount of $^{15}$N-labeled

ribosomes was added to each fraction prior to trypsin digestion. The resulting combination of $^{14}$N- and $^{15}$N-labeled peptides was quantified by LC-MS with the $^{15}$N-labeled peptides used as a reference. Peptides were detected for all r-proteins with the exception of later binding proteins with very low abundance in fraction 1 (S2, S3, S13, S19, S21) and S17 for fractions 1, 4 and 5 (*Figure 1C*). The isotope distribution fits for S17 peptides are often poor for both the experimental and reference sample, preventing unambiguous assignment of these peptides and necessitating their exclusion from the qMS analysis for some fractions. For convenience, the abundance of each r-protein in the experimental sample was normalized to that of the early binder S4, which is expected to be present in all assembling 30S particles. R-proteins were then grouped by the profile of their relative abundances using hierarchal clustering. The protein abundance data is consistent with the Nomura map and previous qMS analysis, with the earliest fractions containing 30S particles with the primary and secondary binders of the 5'- and central domain bound (*Figure 1A,C*) (*Chen and Williamson, 2013*). In contrast, most tertiary binders of the 5'- and central domain (S5, S11, S12) and most 3'-domain binders (S7, S9, S10, S13, S14, S19) are only abundant in later fractions. Moreover, tertiary central and 3'-domain binders (S2, S3, S21) are the least abundant r-proteins in 30S particles across all fractions.

In parallel, fractions from the sucrose gradient were prepared for EM analysis. Images of negatively stained sample were collected for each fraction using automated methods (*Suloway et al., 2005*; *Yoshioka et al., 2007*) and analyzed using single-particle methods (*Lander et al., 2009*; *Mulder et al., 2010*). A mixture of 30S intermediate particles and other abundant large cellular complexes, such as GroEL, is readily observable in raw images collected for the sucrose gradient fractions (*Figure 1—figure supplement 1A*). Fraction 1 contained a particularly low abundance of 30S particles relative to other complexes, and as a result this fraction was omitted from further EM analysis. Given the heterogeneity of particles observed in the raw images, a reference free Difference of Gaussian particle picking method was used to select particles with diameters ranging from 100–300 Å (*Voss et al., 2009*) (*Figure 1—figure supplement 1B*). Particles were aligned and classified using iterative rounds of reference-free alignment, removing non-ribosomal particles between each iteration (*Figure 1—figure supplement 1C*, also See 'Materials and methods' section). The final set of 60 class averages were compared using hierarchical clustering, revealing five major groups of 30S particles at various stages of assembly (*Figure 1B*, *Figure 1—figure supplement 2*). Four of these Groups (I, III, IV and V) resemble the four groups characterized in previous time-resolved EM studies of in vitro 30S reconstitution (*Mulder et al., 2010*), while the Group II class averages were observed in the previous study, but were uncharacterized. In addition to this fraction-by-fraction two-dimensional analysis, random conical tilt (RCT) analysis was performed for fractions from the center of the 30S peak (fractions 3–4), enabling the reconstruction of 3D maps (*Yoshioka et al., 2007*; *Voss et al., 2010*) from representative 2D class averages from each Group (*Figure 2*). Three-dimensional volumes provided additional insight into the stage of assembly of each conformation observed in the 2D class averages.

Consistent with the qMS data, the majority of early assembly intermediates (Groups I–III) observed across the 30S peak are present in the early fractions, with a gradual build up of late intermediates and mature subunits (Groups IV–V) toward the end of the peak (*Figure 1D*). The predominant conformation (Group I) present in early 30S peak fraction 2 is the earliest identifiable intermediate, encompassing the 5'-body and platform regions, but wholly lacking density for the 3'-head domain (*Figure 1D*, *Figure 2A*). The abundance of Group I particles is consistent with the depletion of 3'-domain proteins (S2, S3, S7, S9, S10, S13, S14, S19) observed by qMS (*Figure 1C*).

Both Groups II and III contain head domain density, although the location of the domain relative to the body/platform regions is strikingly different in the two conformations (*Figure 1D*, *Figure 2B,C*). In Group II, the head appears to be completely unanchored from the platform domain, swinging well away from the helix 23/S11 interaction region (*Figure 2B*). In contrast, in Group III the head appears to be docked along the platform domain as it is in mature subunits, but angled away from its final resting spot along the body domain (*Figure 2C*). The vastly different locations of the head domain in Group II and Group III can be readily observed in RCT reconstructions of classes from these Groups (*Figure 2B,C*). RCT volumes for classes from both Groups suggest that late-binding proteins S2 and S3 may be missing from these particles, consistent with the relatively low levels of the proteins observed by qMS (*Figure 1C*, *Figure 2B,C*). In the three RCT reconstructions obtained for classes from WT Group II, the location of the head varies in relation to the 5'-domain. Similarly, the location of head density varies substantially in class averages assigned to Group III, as revealed by focused 2D classification of the head region using custom masks created with Maskiton (*Video 1*) (*Yoshioka et al., 2013*).

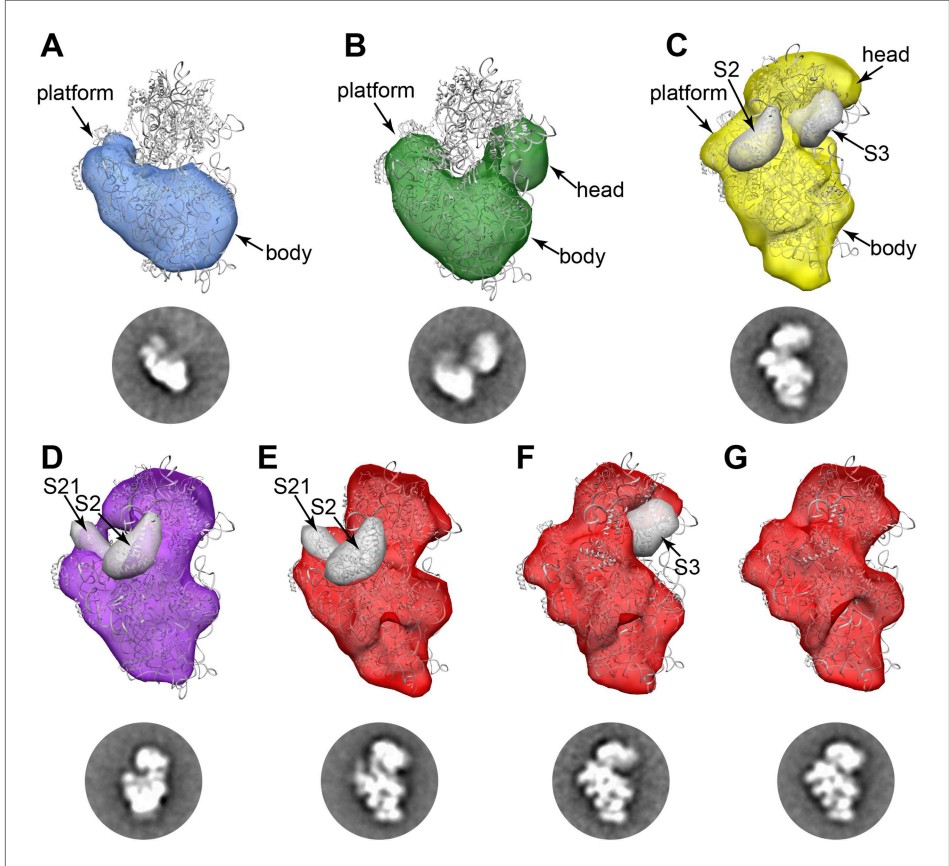

**Figure 2**. RCT reconstructions of assembly intermediates from WT *E. coli*. Representative RCT reconstructions aligned with crystal structure of the mature 30S subunit (PDB 2AVY (*Schuwirth et al., 2005*) shown in gray). (**A**) Group I intermediate. All density for 3′-domain is missing. (**B**) Group II intermediate. Head domain density is detached from the body/platform domain. (**C**) Group III intermediate. The head density is angled away from the body domain, and density for S2 and S3 are missing. A 30 Å filter was applied to the PDB chains for S2 and S3, and the resulting volumes (gray surfaces) lie outside of the RCT volume. (**D**) Group IV intermediate. S2 and S21 density is missing, and the PDB chains for these proteins are shown as gray 30 Å filtered maps as in (**C**). (**E**) Group V intermediate missing S2 and S21, with the PDB chains for these proteins shown as gray 30 Å filtered maps as in (**C**). (**F**) Group V intermediate missing S3 density. The PDB chain for S3 is shown as a gray 30 Å filtered map as in (**C**). (**G**) Fully mature Group V.

This type of 'hinged' head movement is similar to conformations observed in in vitro assembly intermediates, as well as intermediates observed in strains lacking the late-acting assembly factor RimM (*Mulder et al., 2010*; *Guo et al., 2013*; *Leong et al., 2013*).

Classes belonging to Groups IV and V represent very late assembly intermediates and mature subunits, as observed by 2D class averages and 3D RCT reconstructions (*Figure 1D*, *Figure 2D–H*). Groups IV and V are highly represented in later fractions from the 30S peak, consistent with qMS data showing nearly stoichiometric levels of most r-proteins in these fractions. RCT reconstructions of Group IV classes appear to lack some density in the platform region, suggesting that these particles may primarily be late intermediates depleted of very late-binders such as S21 and S2 (*Figure 2D*). Similarly, for Group V, one subgroup of classes that appears to be missing density in the same region of the platform was detected (*Figure 2E*). A second subgroup of classes clearly missing density for S3 but containing density for S2 and S21 was observed (*Figure 2F*), in agreement with previous in vitro observations showing that S2 can bind prior to S3 (*Mulder et al., 2010*). Other Group V particles appear to be fully mature (*Figure 2G*), suggesting that 70S ribosomes may have disassociated during sample preparation. Together, these late-intermediate classes account for the depletion of S2, S3 and S21 observed in the qMS analysis of the late fractions of the 30S peak.

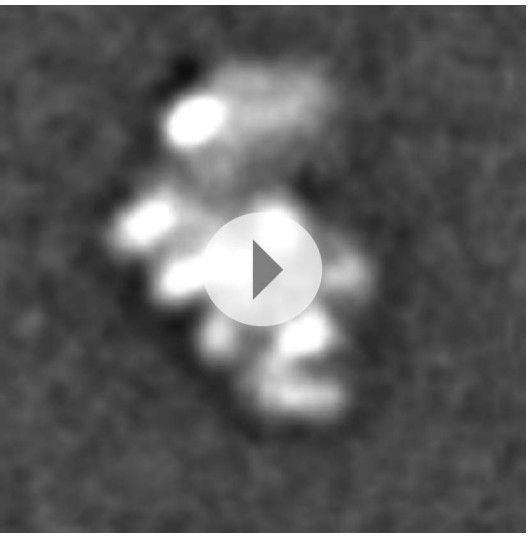

**Video 1**. Analysis of Group III head density movement using Maskiton. A total of 3490 Group III particles were aligned to a reference image using SPIDER (**Frank et al., 1996**). The aligned stack was uploaded to the Maskiton server (www.maskiton.scripps.edu, [**Yoshioka et al., 2013**]), and the Maskiton web interface was used to apply a mask to the head region of the averaged stack. Classifications of the masked region were run for 1000 iterations. The resulting 16 class average images were compiled into a movie using QuickTime Pro 7 (Apple). DOI: 10.7554/eLife.04491.007

## Destabilization of the central pseudoknot region

Among the intermediates observed in WT *E. coli,* the class averages present in Group II were the most intriguing. The vastly different location of the head in these particles compared to Group III particles suggests two alternate pathways for formation of the 3′-domain, either before or after docking of the head onto the platform region of the central domain. In 16S rRNA, the central domain (16S nt:567-912) and 3′-major domain comprising the head (16S nt:920-1396) are connected by a short linker (16S nt:913-919), which forms a long range pseudoknot interaction with helix 1 (h1) at the 5′-end (*Figure 3A,B*). This central pseudoknot (PK) is formed by helix 2 (h2), and is the primary structural feature that leads to the positioning of the head domain upon the platform domain (*Figure 3A,B*). The detachment of the head from the platform in Group II classes suggests that these particles may have unstable or unformed central PK regions. To probe the stability of the RNA in the central PK region, an RNase H cleavage assay was developed using a DNA oligonucleotide anti-sense to the 3′-end of helix 27 (h27) and the 3′-strand of h2 (16S nt:906-920, *Figure 3A*). This 'anti-PK' oligo was incubated with samples encompassing the 30S sucrose gradient peak, allowing the oligo to specifically anneal to assembly intermediates in which the central PK is unformed. The hybridized RNA was subsequently digested with RNase H, resulting in two 16S products encompassing the 5′- (~900 nt) and 3′-domains (~600 nt) (*Figure 3—figure supplement 1*).

Previously, several 30S assembly factors have been implicated in central PK formation providing a motivation for comparing the PK accessibility for both WT *E. coli* and deletion strains using the RNase H assay. The level of central PK formation was determined in *E. coli* BW-25113 and deletion strains of five assembly factors: RimM, RbfA, RimP, RsgA (YjeQ) and KsgA. Alongside these strains, the effects of the plasmid pU23 (*Dammel and Noller, 1993*), that contains a copy of the rRNA operon *rrnB* bearing a C23U mutation within h1 of 16S RNA, was determined. This mutation destabilizes h1 and favors formation of an alternate stem-loop structure within the leader sequence of the pre-cleavage 17S rRNA, thus disrupting formation of h2. When transformed into *E. coli* strain BW-25113, pU23 displays a similar dominant negative cold-sensitive phenotype observed previously in *E. coli* strain DH1 (*Dammel and Noller, 1993*). Each strain was grown at 37°C and harvested during exponential growth, and the 30S peak from the sucrose gradient of each lysate was collected for analysis (*Figure 3—figure supplement 1A*).

RNase H cleavage in the presence of the anti-PK oligo led to accumulation of two products of the expected size (*Figure 3—figure supplement 1B*). Very little product was observed for WT BW-25113, indicating that the majority of intermediates contain fully formed and stable central PKs in this strain (*Figure 3C*, *Figure 3—figure supplement 1B,C*). In contrast, BW-25113+pU23 and all deletion strains displayed substantial but variable amounts of cleavage, suggesting varying degrees of central PK exposure (*Figure 3C*, *Figure 3—figure supplement 1B,C*). Among the deletion strains, only *E. coli* lacking RimP displayed greater cleavage than BW-25113+pU23, suggesting that the Δ*rimP* strain may have the strongest defect in central PK formation. Consistent with a potential role in PK formation, RimP has previously been shown to accelerate binding of the central PK-associated r-proteins S5 and S12 in in vitro 30S assembly assays (*Bunner et al., 2010b*). Negative stain EM datasets for each of the strains were collected in parallel to the RNase H assay, in order to observe the extent of Group II particle accumulation across the 30S peak (*Figure 3—figure supplement 2*). Group II particles were most abundant in Δ*rimP* and BW-25113+pU23 (*Figure 3C*, *Figure 3—figure supplement 2*), and overall the

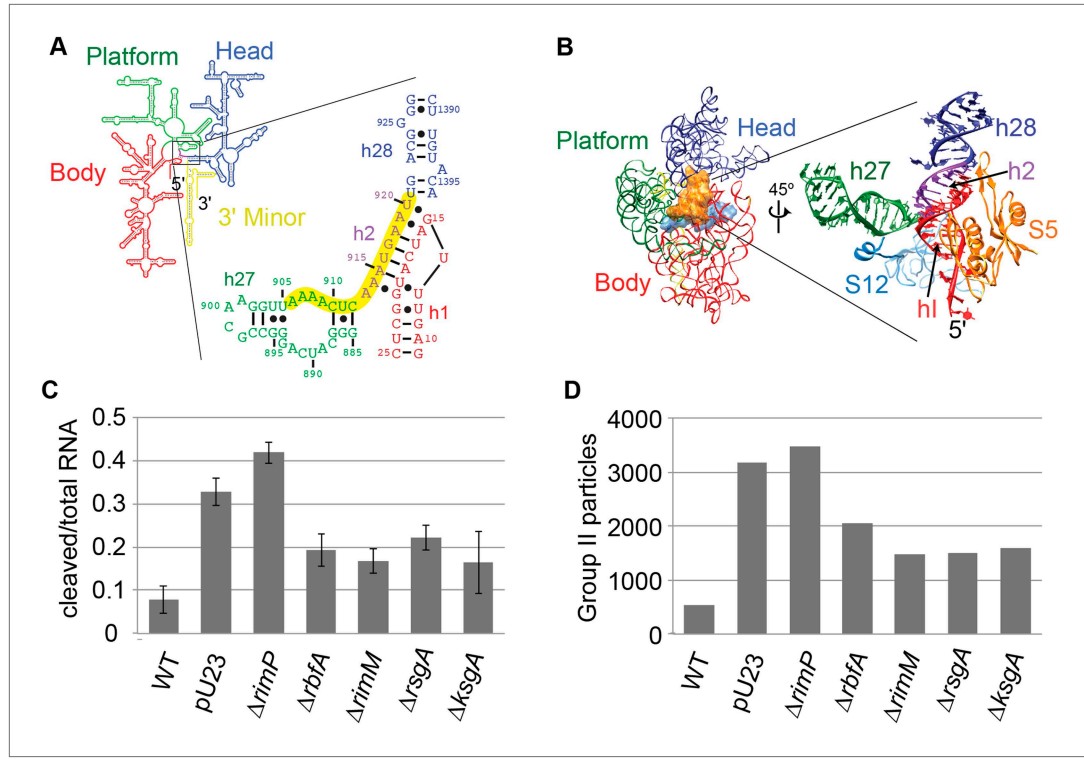

**Figure 3**. Survey of central pseudoknot stability upon deletion of assembly factors. (**A**) 16S rRNA secondary structure (red: 5'-body; green: central; blue: 3'-head; yellow: 3'-minor), with central PK region boxed. The sequences of helix 2 (h2), which forms the central PK, and adjacent secondary structures helix 1 (h1), helix 27 (h27) and helix 28 (h28) are shown at right. The sequence targeted by an anti-sense DNA oligo (16S rRNA nt 906-920) is highlighted in yellow. (**B**) Crystal structure (PDB: 2AVY (**Schuwirth et al., 2005**)) of the 30S subunit from *E. coli*. 16S rRNA is shown as backbone ribbon and colored as in (**A**). Only r-proteins S5 (orange) and S12 (blue) are shown. A close-up of the central pseudoknot and adjacent rRNA helices and r-proteins is shown at right. (**C**) Anti-PK hybridization (500 pmol oligo)/RNase H cleavage of 16S rRNA from 30S peak sucrose gradient fractions for seven different *E. coli* strains. The average fraction of rRNA cleaved (product/total RNA) from three replicates is plotted. Error bars represent the standard deviation of fraction cleaved between the three replicates. (**D**) Abundance of Group II particles in seven *E. coli* strains as measured by negative stain EM. 10000 30S assembly intermediate particles from each strain were combined into a single stack of 70,000 particles. The stack was subjected to reference-free maximum likelihood alignment (**Figure 3—figure supplement 2**). The number of particles from each strain contributing to Group II classes is plotted in the histogram.

The following figure supplements are available for figure 3:

**Figure supplement 1**. Oligo hybridization/RNase H assay.

**Figure supplement 2**. Class averages from seven *E. coli* strains.

levels of particles belonging to Group II in each strain was consistent with the amount of cleavage observed for anti-PK-oligo-dependent RNase H cleavage (**Figure 3C,D**). Based on these results, the Δ*rimP* strain was chosen for further investigation into the composition and conformation of Group II assembly intermediates.

## R-protein depletion in 30S assembly intermediates from Δ*rimP* strain

In order to determine the effect of RimP deletion on the abundance of specific r-proteins in assembling 30S particles, cell lysate from WT and Δ*rimP* strains were prepared for qMS analysis. WT and Δ*rimP* cells were grown in ¹⁴N- and 50% ¹⁵N-labeled M9 media respectively. Equivalent amounts of lysates from each strain were purified using sucrose gradient centrifugation. In agreement with previous studies, the Δ*rimP* strain shows an increase in 30S and 50S particles with a concomitant decrease in 70S particles, compared to the WT strain (**Figure 4A**) (**Nord et al., 2009**). To directly compare the

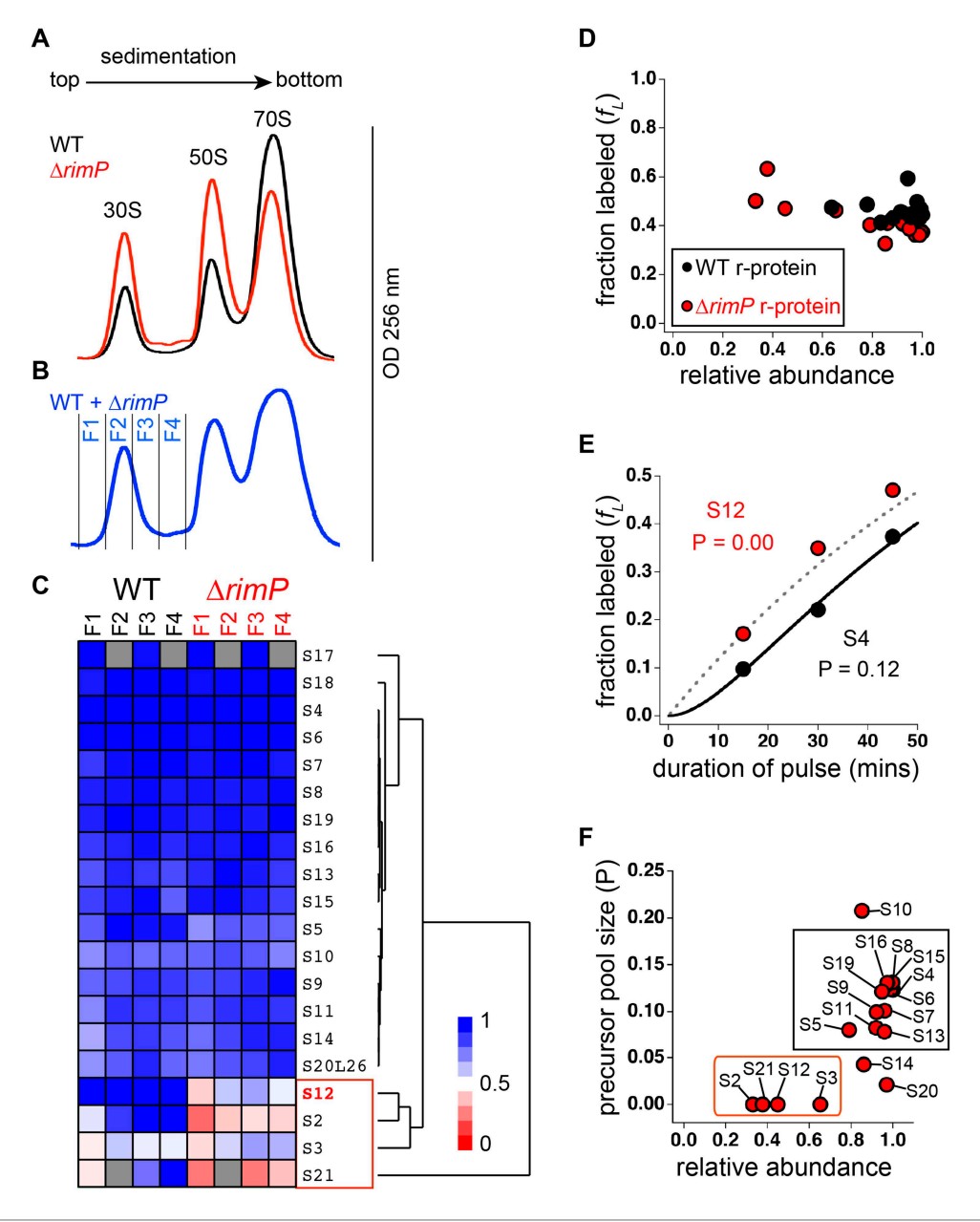

**Figure 4**. Comparison of 30S assembly in WT and Δ*rimP* by qMS. (**A**) Overlay of sucrose gradient chromatograms (absorbance at 254 nm) for WT (black) and Δ*rimP* (red). (**B**) Sucrose gradient chromatogram of combined WT and Δ*rimP* (blue) lysates, with fractions analyzed by qMS labeled 1–4. (**C**) R-proteins in 30S particles in WT (black) and Δ*rimP* (red), across fractions 1–4 (labeled at top) clustered by relative abundance, with a red to blue gradient representing high to low relative abundance. Relative abundance of each r-protein was normalized to that of S4. Gray boxes indicate r-proteins for which no peptides were detected. The cluster comprising r-proteins that are the least abundant in both strains (S3 and S21) and preferentially depleted in Δ*rimP* (S2 and S12) is highlighted by a red box. (**D**) Fraction labeling of 30S r-proteins in 70S particles versus their relative abundance in 30S particles in WT (black) compared to Δ*rimP* (red). Data collected from cells labeled for 45 min. (**E**) Representative labeling kinetics for an early binder, S4 (black) compared to late binder, S12 (red) in Δ*rimP*. The maximum expected labeling rate is represented by the dashed grey line. Time course of experiment was fit (bold black line for S4) to a previously reported pulse-labeling model to determine the precursor pool size (P) of each r-protein (***Chen et al., 2012***). (**F**) Precursor pool size compared to relative abundance of r-proteins in 30S assembly intermediates in Δ*rimP*.
*Figure 4. Continued on next page*

*Figure 4. Continued*

R-proteins with large pool sizes and high abundance in 30S assembly intermediates are boxed in black while those with small pool sizes and low abundance in 30S assembly intermediates are boxed in red.
The following figure supplement is available for figure 4:

**Figure supplement 1**. Pulse labeling experiment to monitor 30S assembly.

abundance of specific r-proteins in assembling 30S particles between the WT and *ΔrimP* strains, equivalent amounts of each cell lysate were combined and purified using sucrose gradient centrifugation (*Figure 4B*). Fractions were collected across the 30S peak and analyzed by qMS as previously described, using $^{15}N$-labeled 70S particles as a reference (*Chen and Williamson, 2013*). Hierarchal clustering of r-protein abundances normalized to that of S4, reveals significant depletion of S3 and S21 relative to other r-proteins in both strains, across all fractions (*Figure 4C*). Furthermore, S2 and S12 are more depleted in the *ΔrimP* strain relative to the WT strain.

The observed incomplete 30S particles could either be degradation products, dead-end assembly intermediates or on-pathway intermediates that eventually mature into 30S and 70S ribosomes. To distinguish among these distinct particle types, we performed a previously described pulse-labeling approach that monitors the flow of $^{15}N$-label during 70S assembly (*Figure 4—figure supplement 1A*) (*Chen et al., 2012*). First, WT and *ΔrimP* cells were pulse-labeled over a time-course at mid-log phase and the 30S particles and 70S ribosomes were purified and analyzed by qMS. The fraction of label incorporated into 70S ribosomes ($f_L$) was determined for each r-protein by comparing the isotope amplitude of the isotope distributions pre-pulse ($^{14}N$) and post-pulse (50% $^{15}N$) (*Figure 4—figure supplement 1B*). Fully assembled 70S particles greatly outnumber assembling 30S intermediates during exponential growth, resulting in faster labeling of the intermediates relative to 70S ribosomes and their degradation products. Therefore, assembly intermediates should have a higher $f_L$ value than 70S ribosomes and their degradation products. The qMS analysis of the fractions containing 30S particles and 70S ribosomes from the WT and *ΔrimP* strains reveals that the 30S particles in the early fractions of both strains have higher $f_L$ values than the 70S ribosomes (*Figure 4—figure supplement 1C*). This indicates that the early fractions of both strains contain significant amounts of 30S assembly intermediates, and that these intermediates are competent to mature into 70S ribosomes.

As assembly progresses, early binding proteins are incorporated into intermediates before late binding proteins, thereby spending more time bound to intermediates than the late binders. Therefore, post-pulse, r-proteins sequentially incorporated into on-pathway intermediates would have different $f_L$ values. Early binding r-proteins would have lower $f_L$ values than late binding r-proteins. Furthermore, any delay in binding of specific r-proteins to on-pathway intermediates would be reflected in their $f_L$ values in 70S ribosomes. By comparing the $f_L$ values of r-proteins in 70S ribosomes in the WT and *ΔrimP* strains to their relative abundance in the 30S intermediates, it can be seen that on average, WT r-proteins are more labeled than those in *ΔrimP* (*Figure 4D*). This indicates that there is a delay in 30S assembly in the *ΔrimP* strain compared to the WT strain, corresponding to an accumulation of assembly intermediates. Moreover, the data show that in the *ΔrimP* strain, the most depleted r-proteins in the 30S intermediates are those with the highest fraction labeled values in the 70S ribosomes, and these correspond to the latest binding r-proteins.

In order to confirm that the incomplete 30S particles are on-pathway assembly intermediates, the *ΔrimP* strain was pulse-labeled and the $f_L$ values of r-proteins in 70S ribosomes were measured for various time periods post-pulse. The $f_L$ values of the r-proteins as a function of the duration of pulse labeling were fit to equations describing the time-course of pulse labeling (*Figure 4E*) (*Chen et al., 2012*). From these fits, the magnitude of the precursor pool size (P) of each r-protein was calculated, reflecting the quantity of unbound r-protein as well as r-protein bound to assembly intermediates. Since dead-end particles do not assemble into 70S ribosomes, they have no effect on the P measured for each r-protein. However, r-proteins in on-pathway intermediates would have P related to their abundance in the precursor pool, including both free protein and assembly intermediates. According to this model, r-proteins that bind early in assembly are the most abundant in intermediates, and are expected to have larger values of P than those that bind later and are less abundant.

The data show that primary binding r-proteins such as S4 that are highly abundant in assembly intermediates in the Δ*rimP* strain have large values of P (p = 0.12 or 12%), confirming that the assembly intermediates are on-pathway (*Figure 4E,F*). Previous studies have shown that WT cells have precursor pool sizes less than 2% (*Chen et al., 2012*). The large precursor pool sizes of early binders in the Δ*rimP* strain are further confirmation of a delay in assembly. In contrast, depleted r-proteins such as S12 have small precursor pools (P ~ 0) (*Figure 4E,F*). The incorporation of these r-proteins is delayed during 30S assembly in the Δ*rimP* strain, and apparently, their synthesis and/or degradation is regulated such that they do not accumulate in their unbound form, resulting in a negligible pool size for these proteins.

The pool sizes for r-proteins in the Δ*rimP* strain cluster into two distinct groups when compared to their relative abundance in the 30S assembly intermediates (*Figure 4F*). One cluster contains r-proteins known to bind early in WT that are highly abundant and have large pools. In contrast, the second group is composed of the latest binders in WT, with one exception, S12. This indicates a marked delay in S12 binding in the Δ*rimP* strain relative to WT. These late binding r-proteins are depleted in intermediates in the Δ*rimP* strain and have small pools. Outliers include S10, which is known to have extra-ribosomal functions (*Friedman et al., 1981*; *Mason and Greenblatt, 1991*), S14, which is exchangeable (*Pulk et al., 2010*) and S20, which is non-stoichiometric (*Hardy, 1975*; *Tal et al., 1990*). Together, these pulse-labeling data show that the intermediate assembly species are on-pathway and that incorporation of S12 and late-binding proteins S2, S21 and S3 is delayed during 30S assembly in the absence of RimP.

## Conformations and distribution of 30S intermediates from ΔrimP strain

Having determined the composition of on-pathway intermediates that accumulate upon deletion of RimP, fractions from the 30S peak of a Δ*rimP* strain sucrose gradient were additionally analyzed by negative stain EM as described above for WT to determine the distribution of conformations present across the gradient (*Figure 5A*). Clustering analysis of class averages from the Δ*rimP* strain revealed five Groups of similar conformations to those observed in the WT dataset (*Figure 5B*). However, the relative number of particles within each Group differs significantly between the two strains, with a dramatic increase in the abundance of Group II intermediates in Δ*rimP* relative to WT (*Figure 1D*, *Figure 5B*). In contrast, the number of particles classified as Group I and III intermediates is similar in the two strains, while the number of late intermediates and mature subunits in Groups IV and V is decreased in Δ*rimP* relative to WT. In order to directly compare the particle conformations and distributions between WT and Δ*rimP* datasets, a combined stack of 10,000 randomly selected particles from each fraction from the two strains was classified using a reference-free maximum-likelihood protocol, followed by clustering analysis of the resulting class averages. For each Group from the cluster analysis, the number of particles contributed from each fraction of either WT or Δ*rimP* was calculated (*Figure 5C*). Similar to the individual fractional analysis, this direct comparison of the combined datasets reveals a substantial relative accumulation of Group II intermediates across the entire 30S peak in Δ*rimP*.

The relative abundance of Group II particles in Δ*rimP* fractions enabled a more in depth analysis of the unusual conformation adopted by this intermediate. In both 2D class averages and 3D RCT volumes, the location of the head volume is highly variable (residues 912–920) (*Figure 5D*). To facilitate the 3D structural analysis, the RCT volumes in *Figure 5D* were aligned based on the body/platform density, and the average density and variance maps between the 10 volumes were calculated (*Figure 5E,F*). The regions of high variance are mainly localized to the head domain, which can sample a substantial range of motion, from locations close to the S11-binding region in the platform domain to the S4-binding region of the body domain. Analysis of Group II particles in 2D by Maskiton (*Yoshioka et al., 2013*) recapitulates this result and additionally indicates that head movement is constrained by a short but highly flexible linker (*Video 2*), likely comprising the 3'-end of helix 27 and the 3'-strand of the unformed h2 (16S residues 910–919). Indeed, the distance between the head and body among Group II RCT volumes is generally 20–40 Å, well within the range of lengths that could be accommodated by a 10-nt ssRNA. PDB models of the 16S rRNA for the body/platform region (nt 1-909) and the head domain (nt 920-1396) were docked into the average density from the 10 RCT volumes (*Figure 5E,G*), and the distance between the two domains could be accounted by the length of the 910–919 linker. A 50-Å filter was additionally applied to the PDB model containing the 16S rRNA and r-proteins (excluding S2), revealing striking similarities to several RCT volumes in the amount of density observed for both the body/platform and head domain (*Figure 5G*). The similarity in size suggests that

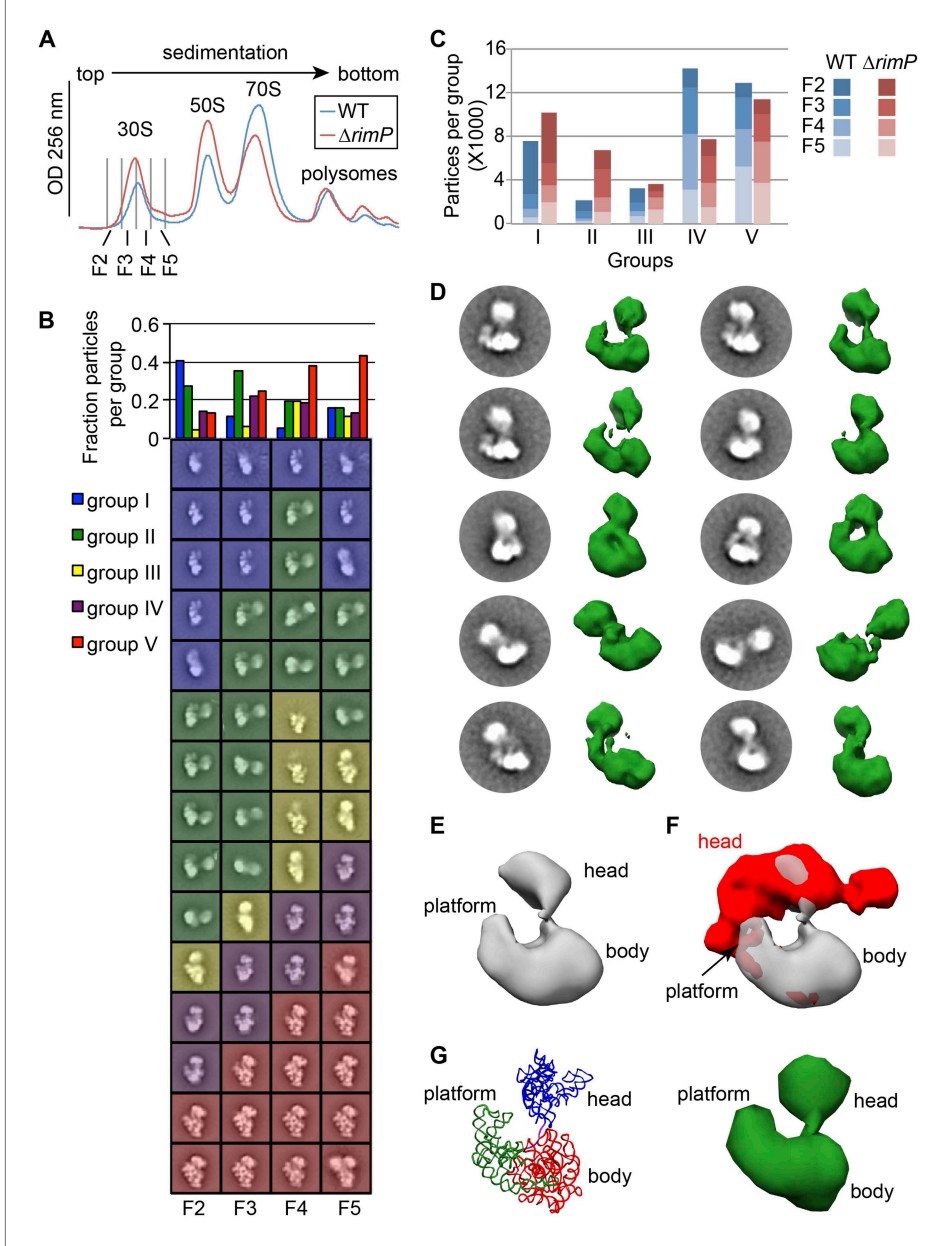

**Figure 5**. Direct comparison of WT and Δ*rimP* assembly intermediates by EM. (**A**) Overlay of sucrose gradient chromatograms (absorbance at 254 nm) for WT (blue) and Δ*rimP* (red) lysates, with 30S peak fractions analyzed by EM indicated. (**B**) Negative stain EM class averages for fractions 2–5 of Δ*rimP* sucrose gradient (labeled at bottom). Classes were obtained by reference-free maximum likelihood alignment and classification and are sorted by Group. Histogram at top shows the fractional contribution of particles from each dataset to each Group. (**C**) Direct comparison of assembly intermediate abundance in WT (shades of blue) and Δ*rimP* (shades of red) strains. 10000 particles from each fraction for each strain were combined into a single stack with 80,000 particles. The stack was subjected to reference-free maximum likelihood alignment. For each strain, the number of particles from each fraction contributing to each Group are plotted as a stacked bar in the histogram, showing the contribution from each fraction and the overall number of particles in each group throughout the 30S peak. (**D**) Two-dimensional class averages and resulting 3D RCT volumes of Group II intermediates from the Δ*rimP* strain. The 3′-head domain location is highly variable between the different volumes. (**E**) Average density of the ten RCT volumes shown in (**D**). (**F**) Variance analysis of the 10 RCT volumes shown in (**D**). The average density from (**E**) is shown in gray, and the variance map is shown in red. (**G**) PDB model of the unanchored head conformation based on location of head in average density of RCT volumes. The 16S rRNA is shown and colored as in *Figure 3A–B*. A 50 Å filter was applied to the PDB, and the density is shown at right.

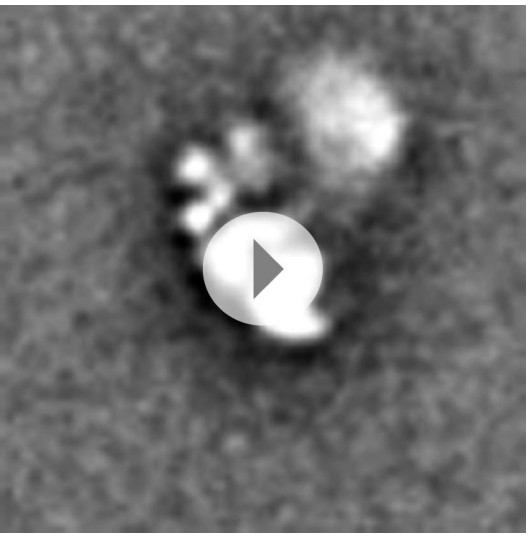

**Video 2**. Analysis of Group II head density movement using Maskiton. Movie was generated as described for **Video 1**, using a total of 3660 Group II particles.

Group II particles may contain a nearly complete complement of r-proteins in the head domain, and that head domain assembly can occur prior to central PK formation.

## Cryo-EM analysis of affinity purified ΔrimP assembly intermediates

The qMS and negative stain EM analysis of Δ*rimP* 30S fractions revealed the r-protein levels and distribution of particle conformations in a complex mixture of various 30S particles and other large complexes. However, the sample complexity prohibited a detailed characterization of assembly intermediates in which the central PK is unformed. To reduce this sample complexity, an affinity purification protocol was developed using a biotinylated oligonucleotide anti-sense to the 3'-strand of h2, similar to the anti-PK oligo used in the RNase H assays described above. This 'capture oligo' was incubated with samples containing 30S subunits and/or intermediates, then bound by NeutrAvidin agarose resin. After thorough washing of the resin, 30S particles annealed to the capture oligo were displaced by adding an excess of a DNA oligo bearing complete complementarity to the capture oligo. Using this purification strategy, 30S intermediates were enriched both from 30S peak sucrose gradient fractions and directly from Δ*rimP* crude lysate (*Figure 6—figure supplement 1A*). In contrast, no 16S rRNA could be detected when purified mature 30S subunits were incubated with the capture oligo (*Figure 6—figure supplement 1A*).

The eluent from the Δ*rimP* intermediate affinity purification was first analyzed by negative stain EM. Whereas samples taken directly from sucrose gradient fractions yielded images containing a large number of non-ribosomal *E. coli* complexes, raw images obtained from the affinity purified sample contained no readily observable non-ribosomal particles, confirming the specific purification of 30S particles. Particle classification further indicated the specific enrichment of early assembly intermediates, with the majority of particles classifying into Group I and II class averages and very few Group III-V classes observed. In addition to the previously identified Groups, an additional class was observed that might be partial degradation products of Group II intermediates corresponding to the 3'-domain of the 30S subunit. Forward projections of the 30S 3'-domain filtered to 30 Å strongly resemble the observed class averages (*Figure 6—figure supplement 1B,C*). In addition, the putative 3'-domain particles varied in abundance based on the amount of 16S rRNA degradation observed in the pull-down sample (*Figure 6—figure supplement 1D*). Together, these observations suggest that particles in these classes contain the final ~600 nt of the 16S rRNA, including the head domain and the 3'-minor domain containing helices 44 (h44) and 45. Indeed, density for h44 could be observed in some negative stain class averages, and was readily observed by cryo-EM (see below, *Figure 6—figure supplement 1C*). The 3'-domain particles appear to be preferentially enriched, suggesting that the 16S:906-920 region is more exposed in these particles than in Group II particles. The 3'-domain particles likely result from non-specific cleavage of the exposed central PK region in Group II particles by contaminating RNases in the sample used for affinity purification. Efforts were made to limit sample degradation using RNase inhibitors, with limited success, further indicating the extent of rRNA exposure in the Δ*rimP* intermediates.

Next, the protein composition of the affinity purified intermediates was analyzed by qMS as described above, using [15]N-labeled 70S particles as a reference. The relative abundance of each r-protein, normalized with respect to S4, shows that S2 and S12 are very depleted in 30S particles with PK instability, with partial depletion of S3 and S5 (*Figure 6A*). The depletion of S2, S3 and S12 in particles with PK instability is consistent with the earlier analysis of all the particles found in the 30S peak. The observed low abundance of S5 in the affinity purified particles could have been masked by the

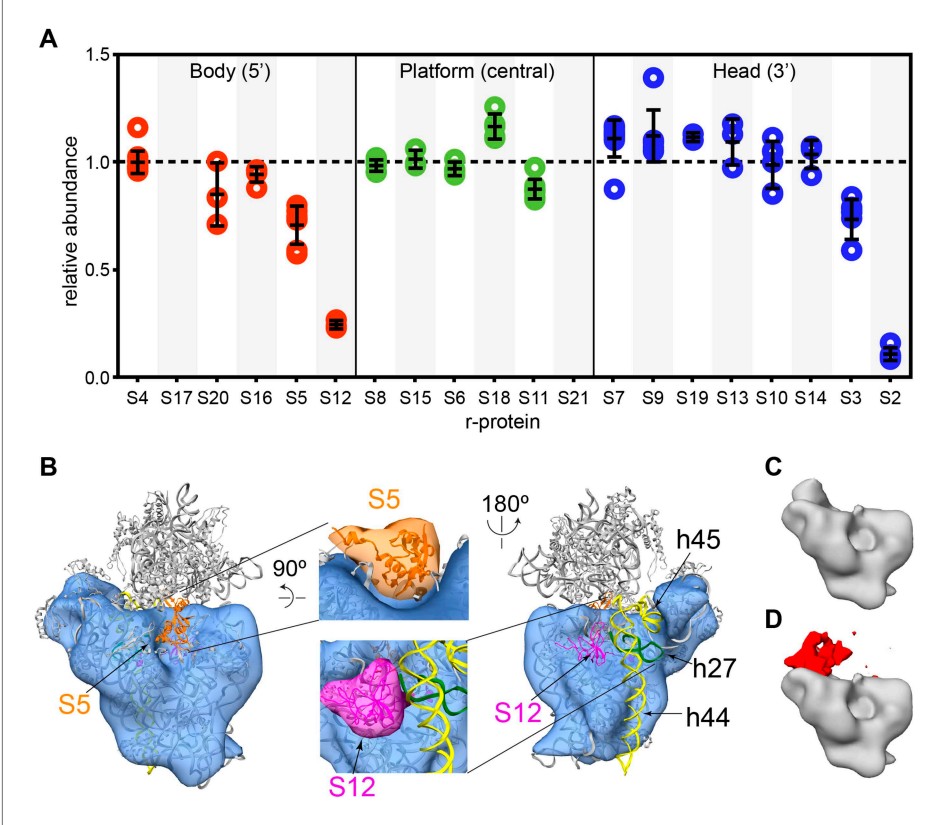

**Figure 6**. Cryo-EM and qMS analysis of affinity purified pre-central PK intermediates. (**A**) Relative abundance of 30S r-proteins grouped by domain bound (body-red, platform-green, head-blue). Relative abundance of each r-protein was normalized to that of S4. No peptides were detected for S17 and S21. (**B**) Representative cryo-EM structure of Group I intermediate. All 3′-domain density is missing, beginning with h27 (green) and continuing through the head and the 3′-minor domain (h44 and h45, yellow). Close-ups of missing body domain r-proteins S5 (orange) and S12 (magenta) are shown at center. The PDB chains for S5 and S12 were filtered to 20 Å, and the resulting maps are located outside of the cryo-EM density. (**C** and **D**) Codimensional PCA variance analysis for Group I cryo-EM particles. (**C**) The average density for all 12,425 Group I cryo-EM particles. (**D**) Variance map for Group I cryo-EM particles (red) overlaid on average map (gray). Regions of high variance are mainly localized in the platform domain. DOI: 10.7554/eLife.04491.015

The following figure supplements are available for figure 6:

**Figure supplement 1**. Affinity purification of pre-central PK intermediates using an anti-PK capture oligo. DOI: 10.7554/eLife.04491.016

**Figure supplement 2**. 3D classification of Group I particles from cryo-EM data set of affinity-purified sample. DOI: 10.7554/eLife.04491.017

presence of a significant amount of particles containing S5 in the untreated sample. Furthermore, the affinity purified particles show a high abundance of most of the 3′-domain r-proteins relative to early 5′-domain binder, S4. This is consistent with the observation that 3′-domain particles are preferentially enriched by the purification procedure. With the exception of S2 and S3, the uniform abundance of all the 3′-domain r-proteins in the purified intermediates suggests that head domain formation is not perturbed until the very late stages of assembly. Some r-proteins, S17 and S21 could not be accurately quantitated in the qMS analysis due to poor fits of their isotope distributions, while the significantly high abundance of S18 is possibly due to its exchange in the 70S particles used as a reference.

The initial negative stain and qMS analysis indicated that purification of 30S intermediates from mature subunits and other *E. coli* complexes substantially reduced sample heterogeneity, allowing for cryo-EM analysis of the affinity-purified sample. Similar to the negative stain analysis, class averages from the cryo-EM data set revealed that the majority of particles are early intermediates (*Figure 6— figure supplement 2A*). Despite efforts to limit sample degradation, a substantial number of 3′-domain

degradation products were observed in the class averages, leading to an increase in the compositional heterogeneity of the data set. In addition, flexibility between the head and body domains in Group II intermediates led to significant conformational heterogeneity for these particles, and limited the ability to reconstruct 3D volumes for these particles. Efforts were therefore concentrated on 3D classification of the Group I cryo-EM particles. Following extensive sorting and classification of Group I particles, four reconstructions of this early intermediate were generated (*Figure 6B*, *Figure 6—figure supplement 2B–D*). As expected, the Group I structures lack density for the entire 3′-domain, including all RNA and r-proteins comprising the head domain and helices 44 and 45 of the 3′-minor domain (*Figure 6B*, *Figure 6—figure supplement 2C*). In addition, 16S rRNA helix 27 lies completely outside the density in all Group I reconstructions, indicating that this final secondary structural element of the central domain is not present in these particles. Notably, the Group I reconstructions also clearly lack density for both S5 and S12, consistent with their observed depletion by qMS. The relatively high levels of S5 in comparison to S12 suggest that S5 may be present in Group II particles, while S12 is likely missing from both intermediates.

Although the four Group I reconstructions are overall very similar in structure, some variability was apparent in the platform region. To identify regions of heterogeneity due to compositional or conformational variability, codimensional principal component analysis (PCA) was performed for the 12,425 Group I particles (*Penczek et al., 2011*) (*Figures 6C,D*). In agreement with variations observed by 3D classification of the particles, this codimensional PCA revealed that variability between these structures mainly arises from differences in the platform region (*Figure 6D*, *Figure 6—figure supplement 2C*). Given the limited resolution of these reconstructions, it is difficult to discern whether these differences are due to compositional or conformational variability in Group I particles. However, the levels of S11 measured by qMS are slightly depleted in comparison to other central domain proteins, in agreement with the variability in S11 density observed in the Group I reconstructions. Together, these analyses of Group I cryo-EM particles suggest that structural stability within the platform region may be dependent on assembly and docking of the 3′-domain.

## Discussion

Over the past 50 years, bacterial ribosome assembly has been studied extensively in vitro using a variety of biochemical and biophysical techniques. These previous studies provided insight into the order, hierarchies and kinetics of r-protein binding and rRNA folding, the fundamental underpinnings of ribosome biogenesis. In contrast, the understanding of in vivo ribosome assembly is relatively modest, owing in part to a lack of tools for the efficient study of this process at a molecular level. Recent developments in biophysical techniques have facilitated more detailed studies into the molecular mechanisms of cellular ribosome biogenesis, and especially the roles of various biogenesis factors. We have developed a high-throughput hybrid qMS/EM approach to study the composition and structure of cellular ribosome assembly intermediates. Our approach allows for the direct comparison of data sets from multiple samples, enabling quantitation of assembly intermediate distribution upon perturbation of the biogenesis pathway.

Both qMS and single-particle EM are ideal methods for the analysis of heterogeneous samples, and both methods were applied to understanding the assembly intermediates present in samples taken directly from a sucrose gradient of crude *E. coli* lysate. Theoretically, all soluble proteins and complexes present in the cell could be observed across the gradient, including all stable ribosomal assembly intermediates. However, other dense cellular complexes that co-elute with 30S assembly intermediates and mature subunits increase sample heterogeneity. Indeed, several abundant proteins and complexes were readily observed in gradient fractions by proteomic analysis and by negative stain EM (*Supplementary file 1*, *Figure 1—figure supplement 1*). At the center of the 30S peak, the majority of the particles (65–70%) were ribosomal; however, fractions on the leading and lagging edges of the 30S peak were far more heterogeneous, containing only 30–35% ribosomal particles. The amount of data required to overcome this sample heterogeneity made high-throughput data collection indispensable for the detection of a wide range of 30S assembly intermediates. The combination of EM and qMS allowed for the classification of these intermediates along the 30S assembly pathway. Moreover, the application of stable isotope pulse-labeling and qMS facilitated the determination of assembly intermediates as on-pathway.

Our analysis of sucrose gradient fractions from the *E. coli* 30S peak revealed five distinct groups of assembly intermediates distinguished by their conformation and r-protein content (*Figure 1C,D*).

Group I particles comprised the earliest observed assembly intermediates with the body and platform domains intact but no head domain density (*Figure 1D*, *Figure 2A*). Group II and III particles all contained head domain density, with the head unanchored from the platform domain in Group II particles and slightly askew in Group III particles, when compared to mature 30S subunits (*Figure 1D*, *Figure 2B,C*, *Figure 4D*). Particles in Group IV and V were the most mature 30S intermediates, containing almost all r-proteins except for those last to be incorporated, namely S2, S3 and S21 (*Figure 1D*, *Figure 2D–H*).

Previous studies have shown that the r-protein binding and rRNA folding can proceed through multiple parallel pathways (*Talkington et al., 2005*; *Mulder et al., 2010*). The difference in the location of head density between Group II and Group III particles suggests that these conformations may result from two such parallel assembly pathways. In particular, the central PK, the long-range tertiary interaction within 16S rRNA that anchors the head domain to the body and platform, appears to be formed in Group III but unformed in Group II particles. Central PK formation occurs late in the assembly pathway (*Powers et al., 1993*; *Besancon and Wagner, 1999*; *Holmes and Culver, 2004*), and could potentially occur after the modular assembly of the 5′-body, central, and 3′-head domains (*Sykes and Williamson, 2009*). This appears to be the case in Group II particles, in which nearly complete density is observed for all three domains in RCT reconstructions (compare *Figures 5D and 5G*). In contrast, Group III particles have partially formed heads that are anchored to the body and platform domains, suggesting that assembly of the head domain proceeds following central PK formation in these intermediates. The presence of both types of intermediates in WT *E. coli* suggests that both pathways are possible in vivo.

A number of assembly factors, including RimM and RbfA, have previously been implicated in central PK formation (*Clatterbuck Soper et al., 2013*). In order to further examine the roles of various factors in this step, we used an anti-PK oligonucleotide hybridation/RNase H assay to test the degree of rRNA exposure in the central PK region. Surprisingly, we found that deletion of RimP has a much stronger effect on central PK exposure than either RimM or RbfA, suggesting that RimP may have a more direct role in central PK formation (*Figure 3B*, *Figure 3—figure supplement 1B,C*). Indeed, EM analysis of intermediates purified from assembly factor deletion strains revealed that Group II particles are most abundant in ΔrimP (*Figure 3C*, *Figure 3—figure supplement 2*). In contrast, ΔrimM predominantly contains Group III intermediates in which the head domain is anchored and only partially formed (*Figure 3—figure supplement 2*), consistent with head-formation defects observed in cryo-EM structures and by in vivo 16S hydroxyl radical footprinting (*Clatterbuck Soper et al., 2013*; *Guo et al., 2013*; *Leong et al., 2013*). RbfA has previously been implicated in the re-structuring of the 5′-leader sequence of 16S rRNA, which must be refolded in order for the central PK to form (*Dammel and Noller, 1995*). Deletion of RbfA leads to defects in folding of the 3′-head domain and the 5′-body domain, including the final placement of h44 (*Clatterbuck Soper et al., 2013*). Our EM analysis of ΔrbfA shows an accumulation of Group II, III and late Group V intermediates (*Figure 3—figure supplement 2*), consistent with the role of RbfA in several stages of assembly. Similarly, ΔrsgA and ΔksgA primarily lead to accumulation of late Group V intermediates (*Figure 3—figure supplement 2*), consistent with previous observations that these factors act in h44 placement during the very late stages of 30S assembly (*Jomaa et al., 2011a*; *Boehringer et al., 2012*).

An in-depth qMS analysis was performed on intermediate 30S particles extracted from the ΔrimP strain and directly compared to those from the WT strain to determine the composition of ΔrimP intermediates relative to those from an unperturbed assembly pathway. The ΔrimP strain was significantly depleted in S2, S3, S12 and S21 when compared to the WT strain. EM analysis of the ΔrimP sucrose gradient fractions revealed that these abundant intermediates are mainly Group I and II particles (*Figure 5B,C*). In particular, Group II is enriched by >3-fold in ΔrimP samples when directly compared to WT (*Figure 5C*). This result suggests that Group II intermediates may either be more long-lived in ΔrimP cells than in WT, or that RimP normally prevents Group II intermediates from forming in WT cells. The ΔrimP intermediates are on-pathway and are eventually incorporated into 70S ribosomes, based on pulse-labeling analysis (*Figure 4E,F*). However, incorporation of depleted proteins occurs at a relatively slow rate, suggesting that the completion of these intermediates is kinetically unfavorable. RimP may act as a chaperone to prevent the assembling 30S subunit from falling into this kinetic trap. RimP has previously been shown to bind to free 30S subunits resolved on sucrose gradients, but not complete 70S ribosomes, indicating that it directly interacts with 30S assembly intermediates (*Nord et al., 2009*).

To directly measure the protein content of pre-central PK intermediates from the Δ*rimP* strain, we devised a strategy for affinity purification using a biotinylated anti-PK oligonucleotide. This early intermediate sample contained relatively low levels of S5 and S3 and significantly depleted levels of S2 and S12 (*Figure 6A*). In addition, the late binding tertiary protein S21 could not be accurately quantitated, likely due to its extremely low levels in the purified intermediates. Cryo-EM structures of the Group I intermediate lack density for all five of these proteins, in addition to all 16S rRNA beginning with h27 (*Figure 6B*, *Figure 6—figure supplement 2C*). It is likely that the severely depleted S2, S12 and S21 are missing from Group II intermediates as well, given that Group II particles are the predominant species in the sample.

The specific depletion of S5 and S12 in early Δ*rimP* intermediates is notable, as both proteins contact the central PK region in mature 30S subunits (*Figure 3B*). Interestingly, RimP has previously been shown to accelerate binding of S5 and S12 by twofold and sixfold, respectively, in in vitro reconstitution assays (*Bunner et al., 2010b*). Together with our findings, these previous results suggest that addition of RimP to 30S reconstitution experiments may help to promote central PK formation and avoid the kinetically unfavorable Group I and Group II intermediates. In previous EM studies of assembly intermediates present during in vitro 30S reconstitution, Group I-like class averages were highly abundant at early time-points during assembly (*Mulder et al., 2010*). Intriguingly, Group II-like classes are also present during the early stages of assembly, although they were uncharacterized in that study (See Figure 1B in *Mulder et al., 2010*). The presence of Group I and II-like classes subsides with the incorporation of S5 and S12, as measured by pulse-chase followed by qMS. These previous in vitro results agree with the present in vivo observations that kinetically unfavorable pre-central PK conformations remain viable on-pathway intermediates.

The central PK is essential for translation (*Brink et al., 1993*; *Poot et al., 1998*) and is conserved in all kingdoms of life. The accurate formation and stability of the central PK is a critical step during small subunit assembly in both prokaryotes and eukaryotes. Recently, the essential ribosomal biogenesis factor Mrd1 was implicated in central PK formation in *Saccharomyces cerevisiae* (*Segerstolpe et al., 2013*). Mrd1 contains multiple RNA-binding domains (RBDs) and binds directly to 18S rRNA helices h27 and h28, two secondary structural elements that reside in close proximity to the central PK in the mature small subunit structure (*Segerstolpe et al., 2013*). Similarly, RimP is composed of two RBDs, and may play an analogous role in binding to 16S rRNA regions adjacent to the central PK. We propose that RimP acts during the early and late stages of 30S subunit biogenesis to assist in the stabilization of the central pseudoknot, allowing for the subsequent incorporation of central-PK binding r-proteins S5 and S12 and late binding r-proteins S2, S3 and S21 (*Figure 7*). During the early stages of 30S biogenesis, premature central PK formation is blocked by a structure within the 16S 5'-leader that is mutually exclusive with h1. RbfA is thought to bind to the leader sequence and promote formation of h1 during the late stages of assembly, and may act synergistically with RimP to stabilize the central PK. In contrast, RimM may act independently of RimP to facilitate assembly of the head domain regardless of the status of central PK formation. Overall, our findings suggest that RimP might be one of the earliest factors to act upon the assembling 30S subunit. The combined EM/qMS approach employed here should have immediate and broad applicability to study of the role of other ribosome assembly factors as well as macromolecular assembly processes involving other bacterial and eukaryotic cellular machines.

## Materials and methods

### Bacterial strains and plasmids

All *E. coli* strains used in this study were in the BW25113 background. BW25113 was used as WT, while assembly factor knockout strains were part of the Keio collection (*Baba et al., 2006*). *E. coli* strains were obtained from either the Yale *E. coli* Genetic Stock Center or Thermo Scientific (Waltham, MA), and genotypes were confirmed by PCR using primers flanking the gene of interest by ~100 bp on either side. The plasmid pU23 encodes a copy of the rrnB rRNA operon containing C23U and C1192U (confers spectinomycin resistance, used for selection purposes in the original study) mutations within the 16S rRNA gene (*Dammel and Noller, 1993*). The plasmid was a generous gift from the Noller lab.

### Bacterial growth and sucrose gradient collection

*E. coli* cultures were grown aerobically in M9 media (glucose, trace vitamins and minerals) supplemented with either $^{14}$N-, 50% $^{15}$N- or $^{15}$N-labeled ammonium sulfate as the only source of nitrogen.

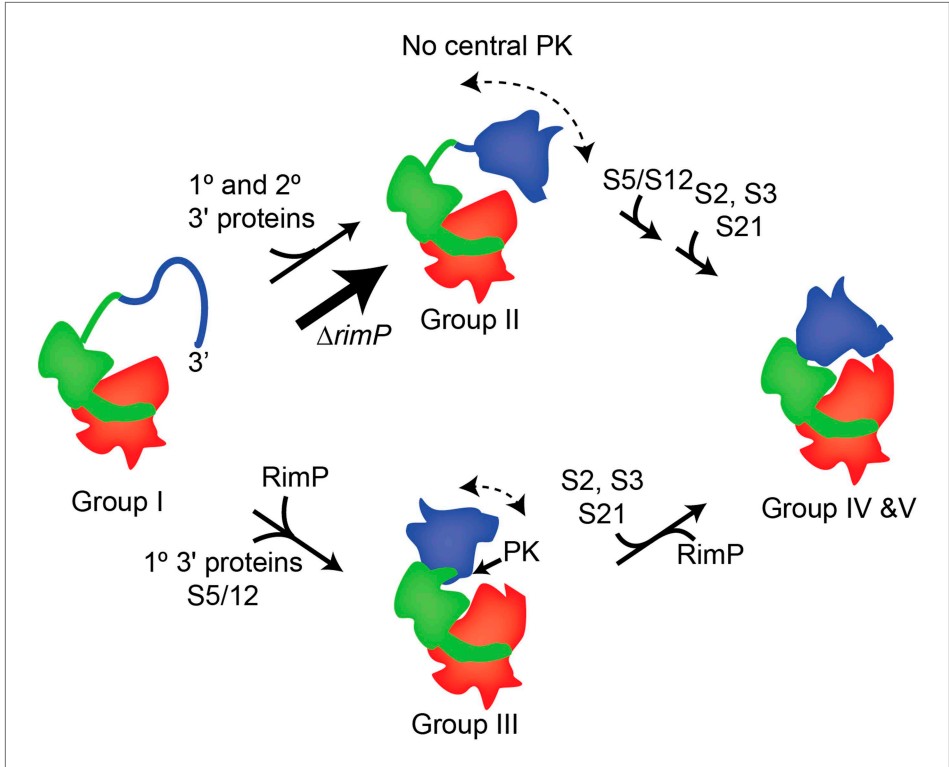

**Figure 7**. A model for 3'-domain formation during *in vivo* 30S biogenesis. Co-transcriptional folding and binding of 5'-body (red) and central domain (green) r-proteins results in the formation and accumulation of Group I intermediates. The 3'-head domain (blue) can fold and r-proteins, including both primary and secondary binders, can bind prior to or following formation of the central PK, resulting in the accumulation of Group II and Group III intermediates, respectively. In the absence of RimP, the central PK is destabilized and the flux of 30S intermediates flows mainly through the Group II pathway, in which the 3' domain is nearly fully formed prior to formation of the central PK. These intermediates are on pathway and eventually all remaining r-proteins, including S5 and S12, are incorporated into the mature 30S subunit.

Cultures were grown to mid-log phase (0.4-0.5 OD600) at 37°C, then quickly cooled by direct addition to ice and harvested by centrifugation (20 min at 6000×g, 4°C). For pulse labeling experiments, cultures were grown to mid-log phase in $^{14}$N M9 media, then pulsed with an equivalent volume of $^{15}$N M9 media for 15, 20, 30 or 45 min. Cells were lysed in Buffer A (20 mM Tris, pH 7.5, 100 mM NH$_4$Cl, 10 mM MgCl$_2$, 0.5 mM EDTA, 6 mM β-Me) by bead beating (0.1 mm Zirconia/Silica beads, BioSpec mini-bead beater), then cleared by two successive centrifugation steps at 22,000×g. The lysate was resolved on a 35 ml 12.9–51.5% sucrose gradient by centrifugation at 4°C in a Beckman SW32 rotor for 16 hr at 26,000 rpm. For samples prepared for qMS analysis comparing the Δ*rimP* and WT strains, equimolar amounts of the respective lysates were combined prior to sucrose gradient centrifugation. The gradients were eluted using a Brandel Gradient Fractionator and prepared for qMS and EM analysis. Fractions containing 70S particles in the WT strain grown in $^{15}$N M9 media were pooled and stored at −80°C for use as a reference sample. Samples used for fraction-by-fraction EM analysis were concentrated to 100 nM if necessary (early and late fractions from 30S peak), and frozen at −80°C. For the assembly factor deletion EM studies, RNase H assays and affinity purification, fractions from the 30S peak were pooled (*Figure 3—figure supplement 1A*), then dialyzed against Buffer A for three hours at 4°C. The samples were concentrated to 250 nM, flash frozen and stored at −80°C.

### Quantitative mass spectrometry analysis

For WT fraction-by-fraction analysis, particles in each fraction under the 30S peak were prepared for LC-MS investigation as previously described (*Chen and Williamson, 2013*). In brief, 20 pmol of each fraction was combined with 20 pmol of 70S spike and trichloroacetic acid (TCA) precipitated

(13% vol/vol final concentration) at 4°C overnight. Precipitates were isolated by centrifugation (30 min at 14,000 rpm at 4°C), washed with 10% TCA, then cold acetone and left to air dry. TCA precipitates were resuspended in 20 µl 100 mM $NH_4HCO_3$, 5% acetonitrile (ACN), 2 µl 50 mM dithiothreitol was added and the mixture was incubated in a 65°C water bath for 10 min. The samples were then treated with 2 µl 100 mM iodoacetamide and incubated for 30 min at 30°C, then digested overnight with 2 µl 0.1 µg trypsin at 37°C. The trypsinized peptides were purified over a Pierce C18 column, eluting across a 5–50% ACN gradient over 105 min. Peptides were then detected on a coupled Agilent G1969A ESI-TOF mass spectrometer over a set detection range of 250–1300 *m/z*. For samples from WT cells only, peptides were detected by an Agilent Q-TOF G6520B and initially processed with Agilent Qualitative Analysis software. Peak lists from the raw LC-MS data were generated using the Aglient Mass Hunter and Mass Profiler programs, and the $^{14}N/^{15}N$ peak pairs were identified and quantified as described previously (*Sperling et al., 2008*; *Sykes et al., 2010*). The relative abundance of each r-protein was then calculated by comparing the amplitude of its $^{14}N$ peptides to that of the sum of its $^{14}N$ and $^{15}N$ peptides [$^{14}N/(^{14}N + ^{15}N)$]. The isotope distribution of each peptide was examined and fits with low signal-to-noise ratios were excluded from further analysis. The relative abundance of each r-protein was normalized to that of the primary binder S4 to flatten any differences in total r-protein amount of each sample. For fraction-by-fraction analysis of the Δ*rimP* strain as compared to the WT strain, qMS analysis was carried out as described above with one exception. In this case, the relative abundance of each r-protein in the Δ*rimP* strain was calculated by comparing the amplitude of the 50% $^{15}N$ peptides to that of the sum of the 50% $^{15}N$ and $^{15}N$ peptides [50% $^{15}N$/(50% $^{15}N$ + $^{15}N$)]. The normalized relative abundance of each r-protein across the 30S peak were hierarchically clustered using Euclidean distance scoring and average linkage in Gene Cluster 3.0. The resulting cluster trees were visualized using Java TreeView.

For proteomic data (*Supplementary file 1*), equimolar amounts of $^{14}N$ peptides were prepared as described above for each sucrose gradient fraction from WT and Δ*rimP* strains. Samples were submitted to an Agilent G6520B QTOF mass spectrometer for LC-MS/MS analysis as previously described (*Chen and Williamson, 2013*). Briefly, peptides were separated by a 90 min 5–60% concave acetonitrile gradient and detected over a precursor detection range of 400 to 2000 *m/z* and a product ion detection range of 80–2000 *m/z*. Data were analyzed using Mascot (precursor mass error tolerance = 0.05 Da, product mass error tolerance = 0.10 Da), and identified peptides were subject to a significance threshold of 0.05 and ion score cutoff of 0.05. The data provided in *Supplementary file 1* represent the highest-scoring match for each peptide.

## Pulse labeling experiments and r-protein labeling kinetics

For pulse-labeling experiments, WT or Δ*rimP* samples were grown in $^{14}N$-labeled media to mid-log phase, then pulsed with an equivalent volume of $^{15}N$-labeled media for 15, 20, 30 or 45 min. At each time-point, 100 ml of culture was rapidly removed and quenched and the cell pellet was stored at −80°C. Time-point samples were then purified by sucrose gradient centrifugation to isolate 30S and 70S particles. Each sample containing either 30S or 70S particles was combined with an equimolar amount of $^{15}N$-labeled 70S particles (reference for accurate peptide identification), and prepared for qMS analysis as described above. For each peptide, the observed raw LC-MS data comprised three isotope distribution envelopes (*Figure 4—figure supplement 1B*). The leftmost (low *m/z*) envelope corresponds to r-proteins synthesized prior to the pulse (100% $^{14}N$) while the middle envelope corresponds to r-proteins synthesized post-pulse (50% $^{15}N$). The rightmost (high *m/z*) envelope corresponds to r-proteins from the reference 70S particles (100% $^{15}N$). The fraction labeled ($f_L$) value of each r-protein was calculated by comparing the abundance of r-proteins synthesized post-pulse to that of the sum of r-proteins pre- and post-pulse [50% $^{15}N$/(100% $^{14}N$ + 50% $^{15}N$)]. For each r-protein, the time course of $^{15}N$-labeling was fit to *Equation (1)* below using Igor Pro (WaveMetrics Inc.) as previously reported (*Chen et al., 2012*).

$$f_L(t) = 1 + P \cdot \exp[-k \cdot (1 + 1/P) \cdot t] - (1 + P) \cdot \exp[-k \cdot t] \tag{1}$$

where $f_L$ is the *fraction labeled* value, P is the precursor pool size, *t* is the length of $^{15}N$ pulse and *k* is the growth rate, with P set as the only free parameter. The curve representing the maximum expected labeling was calculated using *Equation (2)*,

$$f_{max}(t) = 1 - \exp[-k \cdot t] \tag{2}$$

## Electron microscopy sample preparation, data collection and processing

For untilted negative stain EM, samples were applied to plasma-cleaned (20 s, Gatan Solarus) carbon-coated copper mesh grids (Ted Pella, Inc.). For RCT negative stain EM and cryo-EM, samples were applied to plasma-cleaned (5 s) C-flat grids (Protochips) coated with a thin (2–5 nm) layer of continuous carbon. Sucrose gradient fraction samples were diluted with Buffer A to a concentration yielding optimal particle distribution and homogeneity on the grid surface, generally to a concentration of ~10 nM (based on absorbance reading at 260 nm of ~0.13 and 30S extinction coefficient of $12.8 \times 10^6$ $M^{-1}cm^{-1}$). The affinity purified sample was diluted with Buffer A 1:5 for negative stain analysis and 1:3 for cryo-EM analysis. Negative stain grids were prepared by applying the sample (3 µl) to the grid for 1 min, then blotting from the side to remove excess sample. The grid was washed immediately with 3 µl Buffer A, then blotted from the side. Concurrent with blotting, 3 µl of fresh 2% uranyl formate was applied to the grid, then blotted from the side. This step was repeated twice, then the grid was allowed to dry for at least 10 min. For cryo-EM grid preparation, 3 µl of sample was applied for 1 min, blotted for 3 s, then plunge-frozen in liquid ethane using a Gatan CP3.

All EM images were collected using Leginon (*Suloway et al., 2005*). Data for WT and Δ*rimP* fractional analysis were acquired using an FEI T12 transmission electron microscope operating at 120 keV and equipped with a Tietz TemCam-F416 4k × 4k CMOS camera. Images were collected at a nominal magnification of 52000× and pixel size of 2.05 Å with a dose of ~30 e$^-$/Å$^2$ and a nominal focus range from 0.8–1.8 µm under focus. Image tilt pairs (−50°/0°) for RCT data were collected at a dose of ~20 e$^-$/Å$^2$ (*Yoshioka et al., 2007*). Data for 30S peak samples from the WT and knockout strains were acquired using a Tecnai F20 Twin transmission electron microscope operating at 200 keV equipped with a Tietz TemCam-F416 4k × 4k CMOS camera. Images were collected at a nominal magnification of 62,000× and a pixel size of 1.36 Å with a dose of ~30 e$^-$/Å$^2$ and a nominal focus range from 0.8–1.8 µm under focus. Cryo-EM data were acquired using a Tecnai F20 Twin transmission electron microscope operating at 200 keV equipped with a Gatan K2 Summit direct detection device. Cryo-EM images were collected at a nominal magnification of 29000× and pixel size of 1.21 Å with a nominal focus range from 2.5–5.0 µm under focus. Images frame sets (1253) were collected for 6 s with a dose of 33.67 e$^-$/Å$^2$ for 30 frames (200 ms each), followed by whole frame alignment as previously described (*Li et al., 2013*).

For negative stain EM data, all image processing was carried out in Appion (*Lander et al., 2009*). The CTF for all images was estimated using CTFFind3 (*Mindell and Grigorieff, 2003*). For all datasets, particle picking was carried out using DoG picker (*Voss et al., 2009*). Parameters were adjusted to ensure that all particles were selected from each image, in order to eliminate particle selection bias based on size in the initial stack. Following particle extraction (with box sizes ranging from 350–380 Å), the initial stack was subjected to a first round of 2D reference-free alignment and classification using Xmipp ML2D to obtain classes with <2000 particles/class (*Scheres et al., 2005a*, *2005b*). This initial alignment allowed for identification and removal of classes lacking any identifiable features (generally false positive particle picks) or clearly identifiable as a non-ribosomal *E. coli* complex. Identification of non-ribosomal complexes was validated by comparison with known structures of the complexes, and by their presence in proteomic analysis of sucrose gradient fractions (*Figure 1—figure supplement 1C*, *Supplementary file 1*). The cleaned stack was then subjected to reference-free alignment and clustering using Xmipp CL2D to obtain classes with <200 particles/class (*Sorzano et al., 2010*). This finer classification revealed additional false positive and non-ribosomal classes, which were removed. A final round of cleaning was implemented following a second ML2D classification (<500 particles/class). The alternating use of CL2D and ML2D strategies revealed additional class averages containing false positive particles or non-ribosomal classes, although these classes were generally low in abundance and population. The final stack was subjected to ML2D classification with the resultant classes shown in *Figure 1C*, *Figure 4B* and *Figure 3—figure supplement 2B*. The classes were aligned to a reference, and the aligned classes were imported into Mathematica (Wolfram Research) for hierarchical clustering analysis. Dendrograms were constructed using agglomerative hierarchical clustering of the class images, using a correlation distance metric and average linkage clustering. For direct comparison of particle distribution between strains, substacks of 10,000 random particles were created for each data set and combined into a single stack. These combined stacks were then subjected to ML2D alignment and classification. The resultant classes were clustered into Groups in Mathematica as described above.

For RCT data sets, particle tilt pairs were identified using TiltPicker (*Voss et al., 2009*). Untilted and tilted particles were extracted (box size 224, pixel size 2.05 Å) into two separate stacks. For the untilted stack, bad particles were identified and removed using an initial ML2D alignment into 100 classes followed by a CL2D alignment into 256 classes. The cleaned untilted stacks were subjected to ML2D alignment into 15 classes, which were subsequently clustered using Mathematica. Substacks were created for every Group based on the clustering, and each substack was aligned using ML2D. RCT volumes were reconstructed from the resultant classes using the Create RCT Volume function in Appion (*Voss et al., 2010*).

For cryo-EM image analysis, CTF estimation and particle picking were carried out as described above. The micrographs were contrast-inverted, then particles were extracted with a box size of 288 pixels at 1.21 Å/pixel. This box size was optimized for Group I particles, but we also performed a parallel analysis with a larger box size of 320 pixels to examine the larger Group II particles. The initial stack was binned by four and subjected to ML2D alignment and classification to obtain 100 classes. False positive peak picks were discarded, and the cleaned stack was aligned and clustered into 128 classes using CL2D for initial evaluation of the conformations and views present in the sample.

Given the heterogeneity of the sample, it was difficult to distinguish between various conformations and views based solely on visual inspection. We therefore employed a sorting algorithm that compared a set of models to our experimental class averages using Xmipp projection-matching refinement (*Sorzano et al., 2004*). Each class average was assigned to one of the models based on the highest correlation value following projection-matching refinement, as previously described in (*Lyumkis et al., 2013b*). The following five models used for projection matching were created from PDB 2AVY (*Schuwirth et al., 2005*) and low pass filtered to 30 Å: the 3′-domain comprising S3, S7, S9, S10, S13, S14, S19, and 16S nt 921-1534; a Group I model comprising S4, S6, S8, S11, S15, S16, S17, S18, S20, S21, and 16S nt 26-909; a Group II model in which the 3′-domain model (with 16S nt 1398-1534 removed) and the Group I model were fit into a representative RCT volume; a late intermediate missing only S2, S3 and S21; and the fully mature 30S subunit. Notably, no class averages generated by CL2D of the cleaned stack were matched with the mature model. Group I classes were identified from the initial ML2D classes using this projection-matching sorting algorithm aided by visual inspection. These particles were subjected to two further rounds of CL2D to remove bad particles, resulting in a final cleaned stack of 12,425 Group I particles. An initial model was generated from the final CL2D classes using the OptiMod common lines/refinement package in Appion (*Lyumkis et al., 2013c*). An initial set of angles was assigned to the Group I particle stack using Xmipp projection-matching refinement. These angles were further refined and particles were classified through 200 rounds into four models using Frealign 9 (*Lyumkis et al., 2013a*). The final distribution of particles and Fourier shell correlations for the 4 models were as follows: Model 1 – 3407 particles, 27.6 Å; Model 2 – 2975 particles, 26.1 Å; Model 3 – 2722 particles, 27.0 Å; Model 4 – 3241 particles, 25.4 Å. Variance analysis for the Group I cryo-EM particles was performed using the codimensional PCA application in SPARX (*Penczek et al., 2011*). All structure figures were created in UCSF Chimera (*Pettersen et al., 2004*).

## RNase H assays

DNA oligonucleotides used for these experiments were designed to anneal to 16S rRNA positions 906–920 (anti-PK 5′-ATTCATTTGAGTTTT-3′) to test for central PK accessibility or 589–603 (anti-h21 5′- ATCTGACTTAACAAA-3′) as a negative control targeting a highly stable region of the 16S rRNA. RNase H assays were performed in Buffer A, with 10 mM DTT substituted for the 6 mM β-Me. Cleavage reactions were initiated on ice by adding 0.5 pmol 30S subunits (final concentration 33 nM) to 50 or 500 pmol (final concentration 3.3 or 33 μM) anti-PK or anti-h21 oligo (or buffer A for mock reactions) and 5U RNase H (New England Biolabs) (or buffer A for mock reactions). Samples were incubated on ice at 4°C for 16 hr, then resolved on a 2% agarose/TAE gel and visualized by ethidium bromide staining. Intact 16S rRNA and cleavage products were quantified using ImageQuant software, with the two cleavage bands treated as a single product. The intensity of 'cleavage products' detected in the mock reaction lane was subtracted from the cleavage band for each reaction containing oligo, to account for background cleavage that may have occured before or during the RNase H reaction. Fraction cleaved was calculated by dividing the volume of the cleavage products by the total RNA in the lane (cleavage products plus uncleaved rRNA). The average of three replicates was plotted with error bars representing the standard deviation between the three replicates.

## Affinity purification of 30S assembly intermediates

Affinity purification protocol was adapted from (*Schnapp et al., 1998*; *Clatterbuck Soper et al., 2013*). Oligonucleotides were designed based on the anti-PK oligo used for the RNase H assay. The capture oligo comprised a 5′-biotin followed by 10 random DNA nucleotides and finally the 2′-O-methylated anti-PK sequence (5′-biotin-CTACAGGTGCAAmAmUmUmCmAmUmUmUmGmAmGmUmUmU-3′), in order to promoted annealing of the anti-PK sequence to the 16S rRNA. The displacement DNA oligonucleotide was completely complementary to the capture oligo (5′-AAACTCAAATGAATTTGCACCTGTAG-3′). Samples used for affinity purification contained 200 pmol of F2-3 from the *ΔrimP* sucrose gradient (30S peak), 20 OD260 of *ΔrimP* lysate, or 200 pmol purified 30S subunits in 350 µl Buffer A. Each sample was incubated with 17.5 µl 1 mg/ml yeast tRNA, 2 µl 100 µM capture anti-PK oligo (200 pmol), and 5 µl RNasin at 30°C for 15 min. NuetrAvadin agarose beads (100 µl per sample) (Thermo Scientific) were blocked with 0.5 mg/ml BSA in Buffer A twice, then washed with Buffer A, and finally incubated at 30°C for 10 min. Samples were added to beads, then incubated at 30°C for 10 min. Samples were then transferred to 4°C, and incubated with rocking for 2 hr. Beads were centrifuged for 5 min at 500×g, and the supernatant was removed. Beads were washed four times at 4°C with Buffer A + 0.01% Nikkol, then twice more at room temperature with 5 min of incubation with rocking for each wash. Samples were eluted by adding 5 pmol of displacement oligo to 150 µl of Buffer A + 0.01% Nikkol. The buffer and beads were incubated with gentle rocking at room temperature for 30 min, then centrifuged for 5 min at 500×g, and the eluent was removed. This elution was repeated up to three times, but very little 16S rRNA was observed in later elution fractions. Samples were visualized on a 2% agarose/TAE gel stained with ethidium bromide. The first elution fraction was aliquoted and flash frozen in liquid nitrogen, then stored at −80°C prior to EM analysis. Samples for qMS were generated in the same manner, except 275 pmol 30S ribosomes, 500 pmol capture oligo and 250 µl NeutrAvidin Agarose beads were used for affinity purification, and 2.5 µM of displacement oligo was used for elution.

## Data deposition

Electron microscopy maps for the 30S ribosomal intermediates have been deposited to the 3D-Electron Micrscopy Data Bank (EMDB http://www.ebi.ac.uk/pdbe/emdb/) EMDB ID code 6125-6145.

## Acknowledgements

We thank Dr Harry Noller and Dr Laura Lancaster for providing the pU23 plasmid, Dr Anchi Cheng, Dr Sargis Dallakyan, and Dr Joseph Davis for technical support, and members of the Williamson lab and Automated Molecular Imaging group for helpful discussions.

## Additional information

### Funding

| Funder | Grant reference number | Author |
| --- | --- | --- |
| National Institute of General Medical Sciences | R37 GM053757 | James R Williamson |
| National Institute of General Medical Sciences | P41 GM103310 | Clinton S Potter, Bridget Carragher |
| Leona M. and Harry B. Helmsley Charitable Trust | #2012-PG-MED002 | Dmitry Lyumkis |

The funders had no role in study design, data collection and interpretation, or the decision to submit the work for publication.

### Author contributions

DGS, Acquisition of EM data, Analysis and interpretation of EM data, Conception and design, Acquisition of data, Analysis and interpretation of data, Drafting or revising the article; CAG, Acquisition of qMS data, Analysis and interpretation of qMS data, Conception and design, Acquisition of data, Analysis and interpretation of data, Drafting or revising the article; DL, Analysis and

interpretation of data, Drafting or revising the article; CSP, Conception and design; BC, Conception and design, Drafting or revising the article; JRW, Conception and design, Analysis and interpretation of data, Drafting or revising the article

# Additional files

## Supplementary file

• Supplemental file 1. Proteomic data for sucrose gradient fractions from *E. coli* BW25113 and *E. coli* BW25113 Δ*rimP*. Gradient fractions are labeled as in *Figure 1B* and *Figure 5A*. Spectral counts for positively identified proteins are reported for each fraction. A brief gene description is given for each protein.

## Major datasets

The following dataset was generated:

| Author(s) | Year | Dataset title | Dataset ID and/or URL | Database, license, and accessibility information |
|---|---|---|---|---|
| Sashital DG, Greeman CA, Lyumkis D, Potter CS, Carragher B, Williamson JR | 2014 | Electron microscopy maps for the 30S ribosomal intermediates | EMDB 6125-6145 | Publicly available at Electron Microscopy Data Bank |

The following previously published dataset was used:

| Author(s) | Year | Dataset title | Dataset ID and/or URL | Database, license, and accessibility information |
|---|---|---|---|---|
| Schuwirth BS, Borovinskaya MA, Hau CW, Zhang W, Vila-Sanjurjo A, Holton JM, Cate JH | 2005 | Crystal structure of the bacterial ribosome from Escherichia coli at 3.5 A resolution | http://www.pdb.org/pdb/explore/explore.do?structureId=2AVY | Publicly available at RCSB Protein Data Bank |

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
