## [Decision Letter]

Thank you for sending your work entitled “A combined quantitative mass spectrometry and
electron microscopy analysis of ribosomal 30S subunit assembly in *E.
coli*” for consideration at *eLife*. Your article has been
favorably evaluated by James Manley (Senior editor), a Reviewing Editor, and two other
reviewers, both of whom, Harry Noller and Peter Moore, have agreed to reveal their
identity.

The Reviewing editor and the reviewers discussed their comments before we reached this
decision, and the Reviewing editor has assembled the following comments to help you
prepare a revised submission.

The two reviewers praised your paper as exceptional. Their assessment is as follows:

1) This paper addresses one of the most complex and challenging problems in biology –
the mechanism of assembly of ribosomes *in vivo*. The authors use a
combined mass spec/EM approach in an impressive technical tour de force. They first
analyze 30S subunit assembly intermediates obtained from sucrose gradient fractions from
wild-type cells, correlating the absence of specific ribosomal proteins with
characteristic features of EM images of immature particles. They then use their approach
to compare wild-type assembly with that of cells carrying mutations that affect 30S
assembly. A major finding is that the ribosome assembly factor rimP appears to be
critical for formation of the central pseudoknot of 16S rRNA and assembly of its
associated ribosomal proteins. This is an outstanding paper that advances our
understanding of *in vivo* ribosome assembly and demonstrates the use of
a powerful new approach to studying the assembly of complex cellular particles.

2) This manuscript reports the outcome of an extensive series of experiments the authors
have performed to characterize the pathway of ribosome assembly in *E. coli in
vivo*. The powerful technology they have used to do this work has been
perfected by them over a period of several years, and is likely to prove useful for
others interested in the assembly of similarly complex macromolecular assemblies. This
study goes a long way towards proving that RimP, a gene product long known to be
involved in ribosome assembly, plays an important role in ensuring that the central
pseudoknot in 16S rRNA forms properly during assembly.

The insights obtained by the authors reveal the unique power of the experimental system
they have devised, and represent a significant advance in our understanding of ribosome
assembly, a physiological process of the utmost importance biologically.

Minor comments:

1) The “degradation” observed may be due to the presence of endogenous RNase H.

2) The color coding used in Figure 1 was
counter-intuitive. It took quite some time to “read” the protein levels as intended. The
authors might want to consider using a more usual “heat map” range (ramping, say, from
blue to red) to more clearly show the abundance values, which are very critical to the
punch line of the paper.

3) I was puzzled by the absence of signal for protein S17, which was only alluded to in
a cryptic comment toward the end. Perhaps this has already been addressed in a previous
paper?

---

## [Author Response]

1) The “degradation” observed may be due to the presence of endogenous RNase H.

This is one possibility, although we observed increasing amounts of degradation over
time and in the absence of RNase inhibitor, suggesting that the degradation was mainly
due to non-specific RNase activity. We have amended the text as follows to explain our
interpretation of the degradation: “The 3’-domain particles likely result from
non-specific cleavage of the exposed central PK region in Group II particles by
contaminating RNases in the sample used for affinity purification. Efforts were made to
limit sample degradation using RNase inhibitors, with limited success, further
indicating the extent of rRNA exposure in the ΔrimP intermediates.”

Figure legend for Figure 6—figure supplement 1
now reads: “In sample 2, degradation was limited by the addition of RNasin (Promega) and
reducing the amount of time for sample preparation.”

2) The color coding used in Figure 1 was counter-intuitive. It took quite some time to “read” the
protein levels as intended. The authors might want to consider using a more usual “heat
map” range (ramping, say, from blue to red) to more clearly show the abundance values,
which are very critical to the punch line of the paper.

We have changed the coloring of Figures 1 and 4 and Figure 4—figure supplement 1
to ramp from red to white to blue (0-1 relative protein levels).

3) I was puzzled by the absence of signal for protein S17, which was only alluded to in
a cryptic comment toward the end. Perhaps this has already been addressed in a previous
paper?

We generally observe very few peptides for S17, and the peptides that we do detect often
have poor isotope distribution fits. The poor fits cannot be attributed to very low
levels of S17 in experimental samples, because the problem occurs for both the
experimental ^14^N peptides from intact ribosomes and the reference
^15^N peptides. In these datasets, we could not unambiguously assign the
poorly fit S17 peptides, and so they were excluded from our analysis. We have added the
following sentences to the text to clarify this lack of data: “Peptides were detected
for all r-proteins with the exception of later binding proteins with very low abundance
in fraction 1 (S2, S3, S13, S19, S21) and S17 for fractions 1, 4 and 5 (Figure 1). The isotope distribution fits for S17
peptides are often poor for both the experimental and reference sample, preventing
unambiguous assignment of these peptides and necessitating their exclusion from the qMS
analysis.”